# When to Forget? Complexity Trade-offs in Machine Unlearning

**Martin Van Waerebeke** [1]  **Marco Lorenzi** [2]  **Giovanni Neglia** [2]  **Kevin Scaman** [1]

## Abstract

Machine Unlearning (MU) aims at removing the influence of specific data points from a trained model, striving to achieve this at a fraction of the cost of full model retraining. In this paper, we analyze the efficiency of unlearning methods and establish the first upper and lower bounds on minimax computation times for this problem, characterizing the performance of the most efficient algorithm against the most difficult objective function. Specifically, for strongly convex objective functions and under the assumption that the forget data is inaccessible to the unlearning method, we provide a phase diagram for the *unlearning complexity ratio*—a novel metric that compares the computational cost of the best unlearning method to full model retraining. The phase diagram reveals three distinct regimes: one where unlearning at a reduced cost is infeasible, another where unlearning is trivial because adding noise suffices, and a third where unlearning achieves significant computational advantages over retraining. These findings highlight the critical role of factors such as data dimensionality, the number of samples to forget, and privacy constraints in determining the practical feasibility of unlearning.

## 1. Introduction

With the widespread collection of personal data, privacy concerns have become more important than ever (Jha et al., 2024; Kiryati and Landau, 2021). As vast amounts of data are gathered, the risk of errors or damaging information being recorded grows, potentially leading to severe consequences in the training of machine learning models. This, in turn, can result in inaccurate predictions, such as incorrect medical treatment plans and prognosis (Navarro et al., 2021; Lawrence et al., 2024). Regulations such as the GDPR

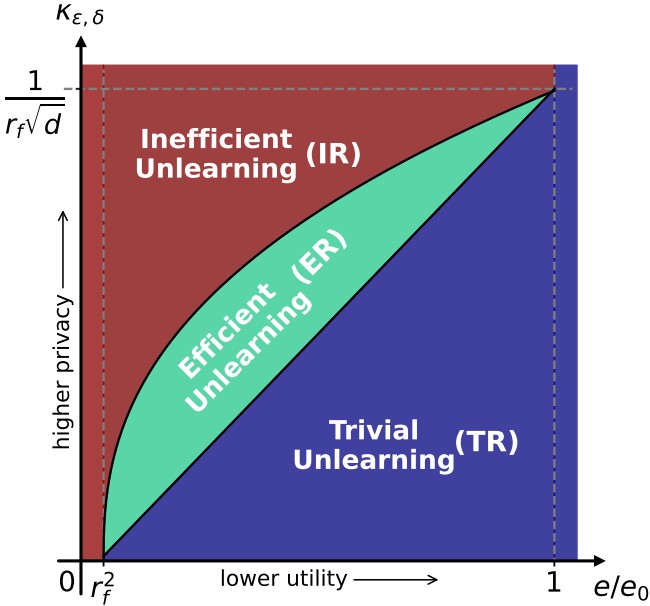

Figure 1: Schematic representation of the phase diagram for unlearning, where $e$ (resp. $e_0$) is the target (resp. initial) excess risk, and $\kappa_{\epsilon,\delta}$ the strength of the privacy constraint (see Section 4). We describe the existence of three regimes of unlearning (IR, ER, TR).

(Mantelero, 2013) and CCPA (Goldman, 2020) attempt to limit these risks and give individuals the right to ask for erasure of their data, but modern machine learning models rely on vast corpora, making it impractical and costly to honor such a request by retraining the model completely (Cottier et al., 2024).

Machine Unlearning (MU) aims to solve this issue by efficiently removing the influence of specific training data (the 'forget set') without the need for full model retraining. MU focuses on developing algorithms that can replicate the effects of retraining from scratch at a "fraction of its cost" (Kong and Chaudhuri, 2023; Georgiev et al., 2024).

While we want to retrain the model *"at a fraction of the cost"*, the exact *"fraction"* is currently not quantified in the literature. Specifically, in a general machine learning setting and for any accepted excess risk $e$, let us define $r_f$ as the fraction of data in the forget set. We denote by

[1] INRIA Paris [2] INRIA Sophia Antipolis. Correspondence to: Martin Van Waerebeke <martin.van-waerebeke@inria.fr>.

*Proceedings of the 42$^{nd}$ International Conference on Machine Learning*, Vancouver, Canada. PMLR 267, 2025. Copyright 2025 by the author(s).

$T_e^U$ (resp. $T_e^S$) the number of stochastic gradient steps required to unlearn the forget set (resp. to retrain from scratch without the forget set) until the excess risk falls below $e$. The efficacy of MU can thus be quantified through the ratio of time required by unlearning over retraining, $T_e^U/T_e^S$, which we call the *unlearning complexity ratio*. We currently lack general bounds on this ratio, making it difficult to rigorously assess the efficiency of MU methods relative to simple retraining.

Our first goal consists in characterizing the regime in which the unlearning complexity ratio is provably lower than one, *i.e.,* when unlearning is possible at a smaller cost than full retraining. The second goal of our work consists in studying the relationship between the unlearning complexity ratio and the amount of data to forget quantified by $r_f$, in order to establish a link between the size of the forget data and the unlearning cost.

**Contributions.** In this paper, we provide a detailed analysis of when unlearning algorithms can lead to efficient data removal. To do so, we provide the first upper and lower bounds on the *unlearning complexity ratio*, and thus characterize the relative performance of unlearning with respect to retraining. In particular:

- We introduce the *unlearning complexity ratio* (unlearning time over retraining time), leveraging minimax optimization complexity in MU for the first time. This allows us to identify regimes where unlearning is inefficient (IR), efficient (ER), or trivial (TR).

- We provide the first lower bound for unlearning complexity, answering an open problem in the literature (Allouah et al., 2024) and showing that there is a regime in which gradient-based unlearning cannot asymptotically beat simple retraining (IR).

- We derive the first upper bound for this ratio, exhibiting a regime in which unlearning is provably faster than retraining (ER). The bound scales with $r_f^2$, the square of the portion of data to forget, demonstrating that a small fraction of data can be unlearned at a small cost.

- We identify a last regime where unlearning is "trivial," as simply adding noise to the parameters suffices, and is thus much more efficient than retraining (TR).

## 2. Related Work

Machine Unlearning is a fast-growing but still relatively new field. Several research directions have emerged in the last years, providing the literature with innovative unlearning methods (Jin et al., 2023; Eldan and Russinovich, 2023) and clever ways of evaluating them (Lynch et al., 2024; Hong et al., 2024). One part of the literature has focused

on efficient methods without certified guarantees (Kurmanji et al., 2024; Eldan and Russinovich, 2023) whose unlearning performance are verified empirically by assessing various metrics such as the accuracy of Membership Inference Attacks (MIAs). In the case of unlearning, the assumption is that the worse the performance of MIAs over the forget set, the better the quality of unlearning. Recently, the efficacy of these methods has been questioned (Aubinais et al., 2023; Hayes et al., 2024). At the same time, the theoretical analysis of MU has known important advances, with new certified unlearning methods (Chourasia and Shah, 2023; Ullah and Arora, 2023; Georgiev et al., 2024), which allow for provable robustness against any MIA.

**Certified Unlearning.** Studies offering certified unlearning guarantees are based on either exact or approximate schemes. While the former offers stronger guarantees by generating models corresponding to exact retraining, they need to modify the training process, usually through some form of sharding (Bourtoule et al., 2021; Yan et al., 2022; Wang et al., 2023), or tree-based approaches (Ullah et al., 2021; Ullah and Arora, 2023). For this reason, in this work we choose to focus on certified approximate unlearning methods, a less explored approach to MU that is steadily gaining attention in the field. Some approximate methods rely on KL-based metrics (Golatkar et al., 2020; Jin et al., 2023; Georgiev et al., 2024), or to linear modeling methods (Izzo et al., 2021) to ensure unlearning. We note that the majority of these approaches provide unlearning certification relying on the theory of Differential Privacy (Guo et al., 2020; Neel et al., 2021; Gupta et al., 2021; Chourasia and Shah, 2023; Allouah et al., 2024).

**Machine Unlearning and Differential Privacy.** Differential Privacy (DP (Dwork and Roth, 2014) ensures that the protected data (the forget set in our case) has a small statistical impact on the final model. One notable advantage of DP-based unlearning certification is its provable robustness against any MIA. Protecting every sample through DP would ensures the privacy of the entire dataset, and thus eliminate the need for unlearning, but would come at the cost of a significant decrease in model utility (Chaudhuri et al., 2011; Abadi et al., 2016). In contrast, in the most common definition of unlearning (Ginart et al., 2019), MU only needs to guarantee the privacy of a small subset of samples, making it a far less restrictive approach. This is emphasized by the fact that several papers in both centralized and decentralized MU offer significant gains in performance and running time as compared to the systematic application of DP (Fraboni et al., 2024; Allouah et al., 2024). The literature of DP-based MU relies on starting from the original optimum, retraining on the retain set (the full dataset minus the forget set) and adding noise to the parameters in order to achieve unlearning. The forget set is thus ignored in order to remove its impact on the weights of the model.

**Comparisons to the Scratch Baseline.** Since the impact of the forget set can be perfectly removed by retraining from scratch, any relevant unlearning method must be faster than simple retraining. This naturally raises the question of how unlearning methods compare with retraining: what computational savings can be achieved, and at what utility cost. While essential, this question is generally challenging to answer due to the complexity of deep-learning models, and the lack of both unified set of hypotheses and unlearning definition. As a result, while it is common to provide an empirical comparison between unlearning and retraining from scratch, this comparison is rarely backed by theoretical arguments. Izzo et al. (2021) provided a comparison of the computational costs of hessian-based MU methods for linear models. Other relevant literature use first-order methods to perform MU. Huang and Canonne (2023) proved that when the retain set is not accessible during unlearning, one cannot outperform DP on the entire dataset. Two recent papers (Chourasia and Shah, 2023; Allouah et al., 2024) have identified upper-bounds on the number of gradient steps needed by their specific MU algorithm to achieve unlearning, and compare these bounds with the ones on full retraining and other methods. Specifically, Chourasia and Shah (2023) consider a stronger definition of unlearning based on adversarial requests and has to rely on the application of DP on the entire training set. (Allouah et al., 2024) is a recent work that considers a setting closer to ours, but focusing however on non-stochastic gradient descent. The authors provide bounds on the number of samples that their two algorithms can delete at a certain utility or computing cost, in contrast to our study of the performance of the best possible algorithms over a wide class.

## 3. Problem Setup

In this section, we describe our learning setup, including the class of iterative learning and unlearning algorithms we consider, and the definition of differential privacy-based unlearning used throughout the paper.

**Notations.** Let $L, \mu \geq 0$, and $\ell : \mathbb{R}^d \to \mathbb{R}$ a differentiable function. We say that $\ell$ is $L$-Lipschitz if, $\forall \boldsymbol{\theta}, \boldsymbol{\theta}' \in \mathbb{R}^d$, $\left| \ell(\boldsymbol{\theta}) - \ell(\boldsymbol{\theta}') \right| \leq L \left\| \boldsymbol{\theta} - \boldsymbol{\theta}' \right\|$, and $\mu$-strongly convex if $\ell(\boldsymbol{\theta}') \geq \ell(\boldsymbol{\theta}) + \langle \nabla \ell(\boldsymbol{\theta}), \boldsymbol{\theta}' - \boldsymbol{\theta} \rangle + \frac{\mu}{2} \left\| \boldsymbol{\theta}' - \boldsymbol{\theta} \right\|^2$. Moreover, we denote as $\left\| \boldsymbol{\theta} \right\| = \sqrt{\sum_i \boldsymbol{\theta}_i^2}$ the $L_2$-norm and as $\mathbb{B}(0, R) = \left\{ \boldsymbol{\theta} \in \mathbb{R}^d : \left\| \boldsymbol{\theta} \right\| \leq R \right\}$ the ball of radius $R$ in $\mathbb{R}^d$. Finally, we will use the notation $\mathrm{supp}(\mathcal{D})$ for the support of a probability distribution $\mathcal{D}$.

### 3.1. Learning and Unlearning Setups

Consider a supervised learning setting in which our objective is to minimize the objective function

$$\min_{\boldsymbol{\theta} \in \mathbb{R}^d} \mathcal{L}(\boldsymbol{\theta}) := \mathbb{E}\left[ \ell(\boldsymbol{\theta}, \xi) \right], \tag{1}$$

where $\ell : \mathbb{R}^d \times \mathbb{R}^s \to \mathbb{R}$ is a loss function, $\boldsymbol{\theta} \in \mathbb{R}^d$ the vector of model parameters, and $\xi \sim \mathcal{D}$ a random data point in $\mathbb{R}^s$ drawn according to a data distribution $\mathcal{D}$. In machine unlearning, our goal is to remove the impact of a subset of the distribution $\mathcal{D}$ from an already learnt model. More precisely, for a fraction $r_f \in [0, 1]$, we decompose our distribution in two parts,

$$\mathcal{D} := r_f \, \mathcal{D}_f \; + \; \left( 1 - r_f \right) \mathcal{D}_r \,, \tag{2}$$

where $\mathcal{D}_f$ is the distribution one wishes to remove (*i.e.,* , the *forget* distribution), while $\mathcal{D}_r$ is the distribution that is unaffected by the removal (*i.e.,* , the *retain* distribution). For example, in order to recover a more common setting over a discrete data set (see *e.g.,* (Ginart et al., 2019)), we may take $\mathcal{D}$ as the uniform distribution over the training dataset $\Xi_N = \{\xi_1, \ldots, \xi_N\}$, $\mathcal{D}_f$ as the uniform distribution over the forget set $\{\xi_1, \ldots, \xi_k\} \subset \Xi_N$, and $r_f = k/N$. Finally, from a model trained on $\mathcal{D}$ (*i.e.,* achieving a small loss $\mathcal{L}(\boldsymbol{\theta})$), our goal is now to recover a good model for $\mathcal{D}_r$ in as little computation time as possible, *i.e.,* a model $\boldsymbol{\theta}'$ whose loss over $\mathcal{D}_r$, $\mathcal{L}_r(\boldsymbol{\theta}') := \mathbb{E}_{\xi \sim \mathcal{D}_r} \left[ \ell(\boldsymbol{\theta}', \xi) \right]$, is small. In order to derive upper and lower bounds for the time complexity of unlearning, we will focus on strongly-convex and Lipschitz optimization problems.

**Assumption 1** (loss regularity). *Let $\mu, L > 0$, and $R = L/2\mu$. For any $\xi \in \mathbb{R}^s$, the loss function $\ell(\cdot, \xi)$ of Eq. (1) is $L$-Lipschitz and $\mu$-strongly convex on $\mathbb{B}(0, R)$.*

Although not directly applicable to most neural networks, these assumptions have already been used in the literature (Sekhari et al., 2021; Huang and Canonne, 2023) and are less restrictive than other recent studies that also rely on smoothness (Chourasia and Shah, 2023; Allouah et al., 2024). We denote as $\mathcal{F}_{sc}(\mu, L)$ the class of such loss functions, abbreviated to $\mathcal{F}_{sc}$ when there is no ambiguity. Moreover, for any function $\ell \in \mathcal{F}_{sc}$, we denote as $\boldsymbol{\theta}^*$ (resp. $\boldsymbol{\theta}_r^*$) the unique minimizer of $\mathcal{L}(\theta) = \mathbb{E}_{\xi \sim \mathcal{D}} \left[ \ell(\boldsymbol{\theta}, \xi) \right]$ (resp. $\mathcal{L}_r(\theta) = \mathbb{E}_{\xi \sim \mathcal{D}_r} \left[ \ell(\boldsymbol{\theta}, \xi) \right]$), and it corresponding loss value $\mathcal{L}^*$ (resp. $\mathcal{L}_r^*$). Note that the choice of radius $R = L/2\mu$ is relatively standard for theoretical analyses (see *e.g.,* Bubeck et al. 2015), as it allows to focus on the regime in which strongly-convex functions offer significantly faster convergence times than their convex counterparts. Finally, following standard terminology in the stochastic optimization literature, we will denote as *computing time* the number of stochastic gradient accesses, or equivalently the number of data samples used throughout the (un)learning procedure.

### 3.2. Iterative First-Order Algorithms

In this, section, we provide precise definitions for learning and unlearning algorithms. More precisely, we will consider that both types of algorithms are *non-deterministic*, *iterative* and *first-order*, *i.e.,* that model parameters are updated through a stochastic iterative procedure that accesses

---

**Algorithm 1** Iterative (Un)Learning Algorithm

**Require:** Update rule $A \in \boldsymbol{A}$, number of iterations $T$, initial model $\boldsymbol{\theta}_0$, loss function $\ell$, dataset $\mathcal{D}$.
1: Initialize memory: $m_0 = \emptyset$
2: **for** $t = 0$ to $T - 1$ **do**
3:      Sample data point: $\xi_t \sim \mathcal{D}$
4:      Compute gradient: $\nabla \ell(\boldsymbol{\theta}_t, \xi_t)$
5:      Update: $(\boldsymbol{\theta}_{t+1}, m_{t+1}) = A(\boldsymbol{\theta}_t, \nabla \ell(\boldsymbol{\theta}_t, \xi_t), m_t, \omega)$
6: **end for**
7: **return** Final model $\boldsymbol{\theta}_T$

---

a stochastic gradient of the loss function at each iteration (see Algorithm 1). This class of algorithms, defined by their *update rule* $A \in \boldsymbol{A}$, is very general and contains most standard optimization algorithms used in machine learning. More precisely, an update rule is a measurable function

$$A\left(\boldsymbol{\theta}_t, \nabla_t, m_t, \omega\right) = \left(\boldsymbol{\theta}_{t+1}, m_{t+1}\right), \qquad (3)$$

where $\boldsymbol{\theta}_t \in \mathbb{R}^d$ is the current model, $\nabla_t \in \mathbb{R}^d$ a stochastic gradient, $m_t \in M$ a memory state, and $\omega \in \Omega$ a seed used for adding randomness into the algorithm. The memory serves as a storage mechanism for essential information about past iterates, enabling the computation of quantities such as momentum, moving averages, or adaptive step-sizes.

For a given update rule $A \in \boldsymbol{A}$, we denote as $\boldsymbol{\theta}_T^A(\boldsymbol{\theta}_0, \ell, \mathcal{D}_r)$ the output of Algorithm 1, which applies the update rule $A$ successively $T$ times, starting at $\boldsymbol{\theta}_0 \in \mathbb{R}^d$.

**Learning Algorithms.** For any update rule $A \in \boldsymbol{A}$, we define the associated *learning algorithm* as the function $\mathcal{A}$ mapping the number of iterations, loss function, and dataset to the output of $A$ initialized at $\boldsymbol{\theta}_0 = 0$, *i.e.,*

$$\mathcal{A}(T, \ell, \mathcal{D}_r) = \boldsymbol{\theta}_T^A(0, \ell, \mathcal{D}_r). \qquad (4)$$

In what follows, we will denote by $\mathbb{A}$ the class of such learning algorithms, and write $\mathcal{A}(\mathcal{D}_r)$ when there is no ambiguity on the values of $T$ and $\ell$.

**Unlearning Algorithms.** While learning algorithms try to estimate the optimum of the objective function $\mathcal{L}_r$ from scratch, unlearning algorithms have the advantage of starting from a pre-trained model with low excess risk (*i.e.,* error of the model minus error of the optimal model) on the whole dataset. More precisely, we will assume that such model was trained for a sufficiently large amount of time, and reached the unique minimizer $\boldsymbol{\theta}^*$ of the objective function $\mathcal{L}$. Therefore, for any update rule $A \in \boldsymbol{A}$, we define the associated *unlearning algorithm* as the function $\mathcal{U}$ mapping the number of iterations, loss function, retain dataset and forget dataset to the output of $A$ initialized at $\boldsymbol{\theta}_0 = \boldsymbol{\theta}^*$, *i.e.,*

$$\mathcal{U}(T, \ell, \mathcal{D}_r, \mathcal{D}_f) = \boldsymbol{\theta}_T^A(\boldsymbol{\theta}^*, \ell, \mathcal{D}_r). \qquad (5)$$

---

**Algorithm 2** "Noise and Fine-Tune" Unlearning Algorithm

**Require:** number of iterations $T$, initial model $\boldsymbol{\theta}^*$, loss function $\ell$, dataset $\mathcal{D}_r$.
1: Sample noise $g \sim \mathcal{N}\left(0, \left(\kappa_{\epsilon,\delta} r_f \frac{L}{\mu}\right)^2 I_d\right)$
2: Initialize model: $\boldsymbol{\theta}_0 = \boldsymbol{\theta}^* + g$
3: Initialize memory: $m_0 = \boldsymbol{\theta}_0$
4: **for** $t = 1$ to $T$ **do**
5:      Sample data point: $\xi_t \sim \mathcal{D}_r$
6:      Compute gradient: $\nabla \ell(\boldsymbol{\theta}_t, \xi_t)$
7:      Update: $\boldsymbol{\theta}_{t+1} = \boldsymbol{\theta}_t - \frac{2}{\mu(t+1)} \nabla \ell(\boldsymbol{\theta}_t, \xi_t)$
8:      Update: $m_{t+1} = m_t + (t+1)\boldsymbol{\theta}_{t+1}$
9: **end for**
10: **return** Final model $\hat{\boldsymbol{\theta}} = \frac{2 m_T}{(T+1)(T+2)}$

---

Again, we will denote as $\mathbb{U}$ the class of such unlearning algorithms, and simply write $\mathcal{U}(\mathcal{D}_r, \mathcal{D}_f)$ when there is no ambiguity on the values of $T$ and $\ell$. Note that these unlearning algorithms can only sample from the retain set to perform unlearning. This is relatively common in the literature of DP-based MU (Neel et al., 2021; Fraboni et al., 2024; Huang and Canonne, 2023; Allouah et al., 2024), although more efficient unlearning methods might exist in scenarios in which the forget dataset is also available during unlearning. Finally, while we allow stateful algorithms in our framework, the algorithm used to achieve our upper bound in Section 4 only uses the state to remember the weighted average of previous iterations rather than all iterations, alleviating some privacy issues for adaptive unlearning requests (Izzo et al., 2021).

### 3.3. Unlearning Guarantees

Unlearning aims at removing the impact of the forget set on the trained model. The way it is achieved in DP-based unlearning is by making sure that the output of the unlearning algorithm is statistically indistinguishable from the output of another algorithm independent from the retain set.

In the literature of DP-based MU, most papers rely on the following definition, first introduced by Ginart et al. (2019).

**Definition 1** (($\epsilon, \delta$)-Reference Unlearning)**.** *An unlearning algorithm* $\mathcal{U} \in \mathbb{U}$ *satisfies* ($\epsilon, \delta$)-*Reference Unlearning if there is a reference algorithm* $\mathcal{A} \in \mathbb{A}$ *such that, for any couple of distributions* $(\mathcal{D}_r, \mathcal{D}_f)$ *and subset* $S \subset \mathbb{R}^d$,

$$\mathbb{P}[\mathcal{U}(\mathcal{D}_r, \mathcal{D}_f) \in S] \leq e^\epsilon \cdot \mathbb{P}[\mathcal{A}(\mathcal{D}_r) \in S] + \delta,$$
$$\mathbb{P}[\mathcal{A}(\mathcal{D}_r) \in S] \leq e^\epsilon \cdot \mathbb{P}[\mathcal{U}(\mathcal{D}_r, \mathcal{D}_f) \in S] + \delta.$$

We refer to it as "reference unlearning" in the sense that an algorithm $\mathcal{U}$ is said to achieve unlearning if another algorithm $\mathcal{A}$ achieves a similar output while being independent from the forget set. However, as mentioned in recent re-

search (Georgiev et al., 2024), this is not satisfying in the sense that it makes the unlearning definition rely on the algorithm $\mathcal{A}$ which is generally simple retraining on the retain set. We propose another -slightly stronger (see Lemma 3.1)- definition that does not rely on a reference algorithm but rather only on the unlearning algorithm itself.

**Definition 2** (($\epsilon, \delta$)-Unlearning)**.** *An unlearning algorithm* $\mathcal{U} \in \mathbb{U}$ *satisfies* ($\epsilon, \delta$)-*Unlearning, if, for any triplet of distributions* $(\mathcal{D}_r, \mathcal{D}_f, \mathcal{D}_f')$, *loss function* $\ell$, *and for any subset of outputs* $S \subset \mathbb{R}^d$, *the following holds,*

$$\mathbb{P}[\mathcal{U}(\mathcal{D}_r, \mathcal{D}_f) \in S] \leq e^\epsilon \cdot \mathbb{P}[\mathcal{U}(\mathcal{D}_r, \mathcal{D}_f') \in S] + \delta.$$

The values ($\epsilon, \delta$) are called the unlearning budget. The lower the budget, the harder unlearning is to achieve.

This definition is very close to the notion of Differential Privacy. The only difference is that this one only ensures privacy of the forget set rather than the entire training set. This definition is very convenient because it does not depend on any other learning algorithm. We will refer to this definition going forward, but note the two are closely related, as proven next.

**Lemma 3.1** (Equivalence of definitions)**.** *Any algorithm achieving* ($\epsilon, \delta$)-*Unlearning also achieves* ($\epsilon, \delta$)-*Reference Unlearning. Furthermore, any algorithm achieving* ($\epsilon, \delta$)-*Reference Unlearning also achieves* ($2\epsilon, (1 + e^\epsilon)\delta$)-*Unlearning.*

Note that the way $\mathbb{U}$ is defined allows non-private algorithms to belong in it. For any unlearning budget ($\epsilon, \delta$), we thus define $\mathbb{U}_{\epsilon,\delta}$ as the algorithms in $\mathbb{U}$ achieving ($\varepsilon, \delta$)-Unlearning.

### 3.4. Minimax Computation Times

In order to quantify whether or not it is worth it to perform unlearning, let us start by introducing some key elements. First, we define the time required to re-learn from scratch and to unlearn. For a given excess risk threshold $e$, loss $\ell$ and learning algorithm $\mathcal{A} \in \mathbb{A}$, one can define the time needed to retrain from scratch until we get an excess risk smaller than $e$ as

$$T_e^S(\ell, \mathcal{A}) := \min_{T \in \mathbb{N}} \{T; \ \mathbb{E}[\mathcal{L}_r(\mathcal{A}(T, \ell, \mathcal{D}_r)) - \mathcal{L}_r^*] \leq e\}, \quad (6)$$

where $\mathcal{L}_r(\boldsymbol{\theta}) = \mathbb{E}_{\xi \sim \mathcal{D}_r}[\ell(\boldsymbol{\theta}, \xi)]$.

In the same way, we can define the minimal number of optimization steps required by an algorithm $\mathcal{U}$ to ($\epsilon, \delta$)-Unlearn the forget set and achieve an expected excess risk under the threshold $e$ starting from the optimum $\boldsymbol{\theta}^*$, *i.e.*,

$$T_e^U(\ell, \mathcal{U}) := \min_{T \in \mathbb{N}} \{T; \ \mathbb{E}[\mathcal{L}_r(\mathcal{U}(T, \ell, \mathcal{D}_r, \mathcal{D}_f)) - \mathcal{L}_r^*] \leq e\}. \quad (7)$$

When studying the performance of an algorithm over a function class, one wants to study the worst-case performance

of any algorithm $\mathcal{A}$ and to find the algorithm minimizing this worst case. Therefore, one can define the minimax re-training time of algorithms in $\mathbb{A}$ over the function class $\mathcal{F}_{sc}$ as

$$T_e^S := \inf_{\mathcal{A} \in \mathbb{A}} \sup_{\ell \in \mathcal{F}_{sc}} T_e^S(\ell, \mathcal{A}). \quad (8)$$

In the same way, we define the minimax forget time of unlearning algorithms in $\mathbb{U}\epsilon, \delta$ over the function class $\mathcal{F}_{sc}$ as

$$T_e^U := \inf_{\mathcal{U} \in \mathbb{U}_{\epsilon,\delta}} \sup_{\ell \in \mathcal{F}_{sc}} T_e^U(\ell, \mathcal{U}). \quad (9)$$

## 4. Regimes of Unlearning Complexity

In this section, we provide lower and upper bounds for the *unlearning complexity ratio* $T_e^U/T_e^S$ (see Section 3.4) over a wide class of unlearning algorithms $\mathbb{U}$. By doing so, we identify regimes in which unlearning methods are significantly faster than retraining, and regimes in which they are not. In particular, we identify values of the target excess risk $e$ and strength of the privacy constraint $\kappa_{\epsilon,\delta} := \epsilon^{-1}\sqrt{2\ln(1.25/\delta)}$ (see the Gaussian mechanism in Dwork and Roth 2014) in which the unlearning complexity ratio is small when forget ratio $r_f$ is small. An illustration of this phase diagram is available in Figure 1, highlighting the three regimes observed in the analysis. All the proofs are deferred to the Appendix.

### 4.1. Speed of Retraining from Scratch

First, we recall the optimal convergence rate for strongly convex and Lipschitz functions, and adapt its proof to our setting. In addition, we show that learning is trivial (*i.e.,* $T_e^S = 0$) if $e \geq e_0 := \frac{L^2}{8\mu}$, as $\boldsymbol{\theta}_0 = 0$ already satisfies the target excess risk.

**Lemma 4.1.** *If $e \geq e_0$, then*

$$T_e^S = T_e^U = 0.$$

In order for our lower bounds to hold, we need a technical assumption that allows flexibility on the choice of forget distribution and objective function, as well as a clear separation between forget and retain distributions.

**Assumption 2.** *(Flexible distributions) For any $p \in [0, 1]$, $\exists A \subset \mathbb{R}^s$ s.t. $\mathbb{P}(\xi_r \in A) = p$, where $\xi_r \sim \mathcal{D}_r$. Moreover, there exists a distribution $\mathcal{D}_f'$ such that $supp(\mathcal{D}_r), supp(\mathcal{D}_f)$ and $supp(\mathcal{D}_f')$ are two-by-two disjoint.*

This assumption is relatively weak, and usually verified for continuous distributions, as long as $supp(\mathcal{D}_r)$ and $supp(\mathcal{D}_f)$ do not cover the whole space $\mathbb{R}^s$.

**Lemma 4.2.** *Under Assumption 2, and if $e < e_0$, we have*

$$T_e^S = \Theta\left(\frac{e_0}{e}\right). \quad (10)$$

To claim that an unlearning method is efficient will thus require for its computation time to be significantly smaller than $O(e_0/e)$. We will show that such unlearning algorithms do exist in Section 4.4.

## 4.2. Trivial Unlearning Regime

We now start with the simplest case: for a high target excess risk $e$ and low privacy constraint $\kappa_{\epsilon,\delta}$, simply adding Gaussian noise to the parameters of the model is sufficient.

**Theorem 1** (Trivial regime). *If the target excess risk verifies* $e \in \left[ \frac{r_f}{1-r_f} \left( \frac{r_f}{1-r_f} + \sqrt{d}\kappa_{\epsilon,\delta} \right) e_0, e_0 \right)$, *then*

$$\frac{T_e^U}{T_e^S} = 0 \,. \tag{11}$$

This first regime corresponds to the dark blue area in Figure 1. In this regime, unlearning can be performed with zero gradient access, provided that the combination of $e$ and $\kappa_{\epsilon,\delta}$ is sufficiently permissive. The boundary of this region, as described in the previous theorem, is an affine function of $e$, as represented in Fig. 1. For low forget ratios $r_f \ll 1$, the regime begins at $e \approx r_f^2 e_0$ and $\kappa_{\epsilon,\delta} = 0$, and ends at $e = e_0$ and $\kappa_{\epsilon,\delta} \approx 1/(r_f\sqrt{d})$. As a direct consequence, for fixed target excess risks $e$ and privacy constraints $\kappa_{\epsilon,\delta}$, unlearning eventually becomes trivial as the forget ratio $r_f$ tends to 0. This result is expected, as the distance bound between the two optima $\|\theta^* - \theta_r^*\|$ is proportional to $r_f$, and thus unlearning algorithms start directly from the optimum in this regime (see Lemma C.1).

## 4.3. Inefficient Unlearning Regime

Conversely, we now show the existence of a regime in which unlearning cannot asymptotically outperform retraining.

**Theorem 2** (Inefficient regime). *Let $\delta \in [10^{-8}, \epsilon]$. Under Assumption 2, there exists a universal constant $c > 0$ such that, if $e < \min \left\{ 1, c \left( \frac{r_f}{1-r_f} \right)^2 \left( 1 + \kappa_{\epsilon,\delta}^2 \right) \right\} e_0$, then*

$$\frac{T_e^U}{T_e^S} = \Omega(1) \,. \tag{12}$$

The proof of Theorem 2 relies on three steps: 1) defining a class of objective functions $\mathcal{L}^g$ for $g : \mathbb{R}^s \to \{-1, 1\}$ such that their optimum over $\mathcal{D}$ does not provide any information on the dataset $\mathcal{D}_r$, 2) showing that two such functions $\mathcal{L}^g$ and $\mathcal{L}^{-g}$ have optimums over $\mathcal{D}_r$ distant from one another, and 3) showing that any algorithm's output will behave nearly identically on both $\mathcal{L}^g$ and $\mathcal{L}^{-g}$, thus leading to the impossibility of having both functions efficiently optimized by the same algorithm. Overall, the approach is similar to Le Cam's two point method (see for instance Polyanskiy and

Wu 2025), but requires combining this classical method with the $(\epsilon, \delta)$-Unlearning constraint in order to derive proximity between the two algorithm's outputs.

Theorem 2 provides a regime in which first-order unlearning methods cannot asymptotically outperform retraining. This regime is delimited by a curve of type $\kappa_{\epsilon,\delta} \geq \alpha\sqrt{e}$, with $\alpha$ a constant, explaining our choice of representation in Fig. 1. For low forget ratios $r_f \ll 1$, the unlearning complexity ratio is lower bounded by a constant when $e$ is below a quantity proportional to $r_f^2(1 + \kappa_{\epsilon,\delta}^2)e_0$. In this regime, removing even minimal parts of a dataset requires a non-negligible retraining time. This may be an issue when numerous small removals must be made to a model, as each of these removals will incur a cost proportional to that of its full retraining. Fortunately, $r_f^2 e_0$ is often extremely small in practical scenarios when the forget ratio is proportional to $1/n$ where $n$ is the size of the training dataset (see Section 4.5), and this regime only appears for very high privacy constraints $\kappa_{\epsilon,\delta}$ of the order of $\frac{1}{r_f}\sqrt{\frac{e}{e_0}}$. Finally, when the target excess risk $e$ tends to 0, a direct corollary of Theorem 2 is that unlearning cannot asymptotically outperform retraining, regardless of the strength of the privacy constraint $\kappa_{\epsilon,\delta}$. However, most existing exact MU algorithms rely on modifications of the training set and are thus not included in our framework.

**Corollary 1.** *Within the hypothesis of Theorem 2, and for $r_f \in (0, 1)$ and $\kappa_{\epsilon,\delta} \geq 0$ fixed, we have*

$$\liminf_{e \to 0} \frac{T_e^U}{T_e^S} > 0 \,. \tag{13}$$

In other words, the advantage of starting from a pre-trained model $\theta^*$ instead of retraining from scratch reduces as the target excess risk decreases, and, below a certain threshold, no asymptotic complexity improvement can be achieved by unlearning methods over retraining. Additionally, Theorem 2 offers an impossibility result for exact unlearning methods within our assumption set. Indeed, exact unlearning implies setting $\kappa_{\epsilon,\delta} = +\infty$, applying Theorem 2 thus informs us that no asymptotic performance gain is possible.

## 4.4. Efficient Unlearning Regime

We have now identified that, on one end of the spectrum, unlearning is trivial, while on another end, unlearning is inefficient. We now characterize what happens between those two extremes by showing that a simple unlearning mechanism achieves a good unlearning complexity ratio in this intermediate regime. To do so, we derive an upper bound on unlearning time using the unlearning algorithm "noise and fine-tune" (see Algorithm 2), which is an adapted version of Neel et al. (2021)'s perturbed gradient descent. Using this upper bound along with Lemma 4.2, we immediately get an upper bound on the unlearning complexity

ratio. Although the choice of this simple algorithm may not be optimal, it only impacts the results in this subsection, whereas Theorems 1 and 2 are agnostic to the choice of the algorithm.

**Theorem 3** (Noise and Fine-Tune Efficiency)**.** *For any $e < e_0$, we have*

$$\frac{T_e^U}{T_e^S} = \mathcal{O}\left( \left( \frac{r_f}{1 - r_f} \right)^2 \left( 1 + d\kappa_{\epsilon,\delta}^2 \right) \frac{e_0}{e} \right). \qquad (14)$$

The proof of Theorem 3 relies on controlling the distance between the two optima $\|\boldsymbol{\theta}^* - \boldsymbol{\theta}_r^*\|$ by a factor proportional to $r_f$, and using the classical convergence rate in $\mathcal{O}(LR/\sqrt{T})$ for convex stochastic optimization when the optimum is within a distance $R$ from the initialization. Quite surprisingly, this convergence rate proportional to $r_f$ is obtained with the use of convergence rates from the convex literature, despite being in a strongly convex regime where faster rates in $\mathcal{O}(L^2/\mu T)$ are usually favored.

Theorem 3 shows that efficient unlearning, with an unlearning complexity ratio proportional to $r_f^2$, is possible. For low forget ratios $r_f \ll 1$, the "noise and fine-tune" method outperforms retraining (*i.e.*, $T_e^U < T_e^S$) when the target excess risk is above a quantity proportional to $r_f^2 \left( 1 + d\kappa_{\epsilon,\delta}^2 \right) e_0$, and we recover, up to a constant and for a fixed dimension $d$, the regime in which unlearning becomes possible in Theorem 2 (see Section 4.3). The combination of both Theorem 2 and Theorem 3 thus shows that $r_f^2 \left( 1 + d\kappa_{\epsilon,\delta}^2 \right) e_0$ acts as a threshold for the target excess risk before which efficient unlearning is impossible, and above which unlearning becomes efficient (and even trivial beyond $r_f(r_f + \sqrt{d}\kappa_{\epsilon,\delta})e_0$).

Moreover, while a large value of $\kappa_{\epsilon,\delta}$ negatively impacts the unlearning time, the converse is only true to a certain extent since a low value will not necessarily allow for immediate unlearning. This is due to the fact that, even with very a weak privacy constraint, the unlearning algorithm still has to cover the distance between the full optimum and the retain optimum.

To complete the characterization of the unlearning regimes initiated in this section and illustrated in Figure 1, we conclude with the following direct corollary of Theorem 3, which characterizes the efficient regime.

**Corollary 2.** *(Efficient regime). There exists a universal constant $c > 0$ such that, for any $\gamma \in (0, 1)$, if*

$$e \geq \frac{c}{\gamma} \left( \frac{r_f}{1 - r_f} \right)^2 \left( 1 + d\kappa_{\epsilon,\delta}^2 \right) e_0,$$

*then*

$$\frac{T_e^U}{T_e^S} < \gamma.$$

### 4.5. Discussion

Overall, our analysis shows that there are three main regimes—Trivial, Inefficient, and Efficient—that describe how unlearning time compares to retraining-from-scratch time, based on the target excess risk $e$, the strength of the privacy constraint $\kappa_{\epsilon,\delta}$ and the forget ratio $r_f$. Figure 1 illustrates these regimes and their boundaries.

Since we rely on noising the model parameters to ensure unlearning, our bound scales with $\sqrt{d}$, as is common in differentially-private optimization (Bassily et al., 2014). While natural, this dependence is not matched by our lower bound in Theorem 2. We leave the exploration of this discrepancy to future work.

In a realistic machine learning setting, one is usually interested in achieving an excess risk on the training set of the same order as the generalization error, which value if often around $1/n$, where $n$ is the size of the dataset (Bousquet and Elisseeff, 2002). In this context, forgetting one sample with an average unlearning budget (*i.e.,* $\kappa_{\epsilon,\delta} \approx 1$) can be done fairly cheaply, since applying the bound from Theorem 2 yields $T_e^U/T_e^S$ of the order of $r_f^2 d \frac{e_0}{e} \approx r_f^2 dn$, and one can thus remove a number of data points at most of the order $\sqrt{\frac{n}{d}}$. Moreover, if $r_f \propto 1/n$, then the unlearning complexity ratio is of the order $d/n$ and unlearning can be very efficient in under-parameterized settings.

## 5. Experiments

We investigate the global landscape of the unlearning complexity ratio as a function of key factors, including the accepted excess risk threshold $e$ and the unlearning budget $(\epsilon, \delta)$, which are jointly quantified by the constant $\kappa_{\epsilon,\delta}$.

### 5.1. Experimental Setting

The goal of the experiment section is to validate the theoretical analysis presented in Section 4 by comparing the performance of unlearning and retraining on both real and synthetic functions and datasets.

In order to give an estimate of the unlearning complexity ratio, we need to choose specific algorithms to represent the learning algorithm class $\mathbb{A}$ as well as the unlearning algorithm class $\mathbb{U}$. For the learning algorithm, we choose stochastic gradient descent, as defined in (Garrigos and Gower, 2023), since it is known to achieve the optimal asymptotic speed of $\Theta(\frac{e_0}{e})$ (see proof of Lemma 4.2).

For the unlearning algorithm, we choose the "noise and fine-tune" algorithm (see Alg. 2) since it is the one used to derive the bound in Theorem 3.

We aim to learn linear regression models in $\mathbb{R}^d$ (with even $d$). We perform experiments both on synthetic "worst-case"

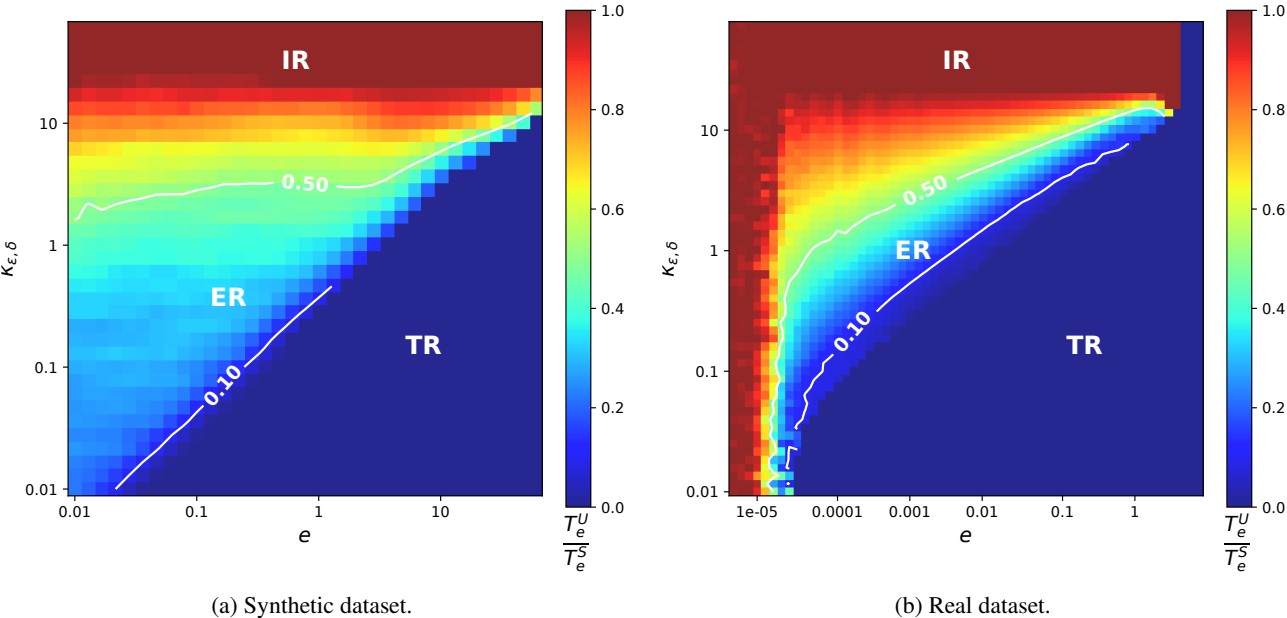

(a) Synthetic dataset.
(b) Real dataset.

Figure 2: Experimental phase diagram of the unlearning complexity ratio. We give estimates for $T_e^U$ and $T_e^S$ using the "noise and fine-tune" (Algorithm 2) and SGD algorithms, respectively. We display the value of their ratio as a function of $\kappa_{\epsilon,\delta}$ and $e$ in log-log scale. We notice the three regimes described in our theoretical analysis: inefficient (IR), efficient (ER), and trivial (TR). .

functions, as analysed in our theory, and on the Digit dataset of handwritten digits, which is a subset of the larger dataset proposed in Alpaydin and Alimoglu (1996).

In every experiment, the retain and forget are obtained through the random splitting of the dataset into two parts of respective sizes $n - \lfloor r_f * n \rfloor$ and $\lfloor r_f * n \rfloor$. We consider $r_f = 10^{-2}$.

### 5.2. Experiments on Synthetic Data

For the synthetic dataset, the expected loss function is:

$$\mathcal{L}(\boldsymbol{\theta}) = \frac{\mu}{2} \|\boldsymbol{\theta}\|^2 - \frac{L}{4} \mathbb{E}_{\xi \sim \mathcal{D}}[g(\xi)] \sum_{i=1}^{d/2} \theta_i + \frac{L}{4} \sum_{i=1+d/2}^{d} |\theta_i|,$$

where $g : \mathbb{R}^s \mapsto \{-1, 1\}$ and $\theta_i$ represents the $i$-th coordinate of $\boldsymbol{\theta}$. The dataset distribution $\mathcal{D}$ influences the loss solely through the resulting distribution of $g(\xi)$, where $g(\xi)$ is a Rademacher random variable with an expected value of $\mathbb{E}[g] = \frac{1}{2\sqrt{T}}$, where $T$ denotes the number of time steps for which the experiment will be conducted. For this experiment, we set $L = 25$ and $\mu = 1$.

This loss function stems from the refinement of the "worst-case" loss analyzed during the proof of Theorem 2, with the goal of deriving a hard function fitting our hypothesis. Its origin is given in more details in Appendix B.

The unlearning and retraining algorithms are performed for a fixed time $T$. To produce the phase diagram of Fig. 2b, we choose values of $T$ logarithmically spread between 1 and $10^6$. We also choose values of $e$ and $\kappa_{\epsilon,\delta}$ logarithmically spread between $10^{-2}$ and $10^2$. For each value of $T$, we generate a corresponding function $g$ and loss function $\mathcal{L}$. We run our learning algorithms for $T$ stochastic gradient steps, recording the corresponding time every error threshold $e$ is passed. For each value of $\kappa_{\epsilon,\delta}$, we run our unlearning algorithm for $T$ stochastic gradient steps, recording the corresponding time when every error threshold $e$ is passed. We repeat this process 50 times to get better estimations of the expected unlearning and retraining times. Then, for each value of $e$, we average all of the obtained learning times. For each pair of value $(e, \kappa_{\epsilon,\delta})$, we average all of the obtained unlearning times. The ratio of the average unlearning time and of the average learning time is what we refer to as the empirical unlearning complexity ratio. By definition, this is set to 0 if the average unlearning time is 0.

### 5.3. Experiments on Real Data

For the real data, the experimental process is simpler as we optimize a standard cross-entropy loss with $L2$ regularization. For various values of $\kappa_{\epsilon,\delta}$ and $e$, we measure $T_e^U$ and $T_e^S$ in a more realistic machine learning setting, with decay-

ing learning rate, batch size of $64$, and averaging the results over $50$ runs. We defer the full experimental details to Appendix D. The sensitivity used for the unlearning algorithm is the real distance $\|\boldsymbol{\theta}^* - \boldsymbol{\theta}_r^*\|$.

## 5.4. Experimental Results

Figure 2 illustrates the empirical unlearning complexity ratio. The figure empirically demonstrates the existence of the different regimes theorized in Sec. 4. More specifically, Theorem 1 predicts an affine linear dependency between $e$ and $\kappa_{\epsilon,\delta}$ at the frontier of the efficient regime and the trivial regime. In the log-log plot, this would appear as a line with slope $1$, which we can recognize in the level-curve corresponding to an empirical complexity ratio equal to $0.1$ in Fig 2a[1]. We can however see that that this is not the case when experimenting on the smooth loss function of Figure 2b, as the smoothness allows for faster convergence rates, and the delimitation thus becomes a square-root function.

The existence of the efficient regime ER can also be noticed in Fig. 2. A large portion of the figure shows values of the experimental unlearning complexity ratio well below $1$. We observe that, while our experiments considered an optimal learning algorithm, the "fine and tune" unlearning algorithm may be sub-optimal. One can expect that better unlearning algorithms can achieve even smaller complexity ratios.

Finally, we also observe the IR regime at the top of the figure. Infact, the low slope of the level line at $0.5$ indicates a behaviour similar to what is theoretically predicted by Theorem 3 as a square root function has a slope of $0.5$ in log-log scale.

## 6. Conclusion

In this paper, we study the efficiency of machine unlearning through the lens of a novel metric —the *unlearning complexity ratio*— which compares the worst-case convergence speeds of the best unlearning and retraining algorithms. Our analysis reveals three distinct regimes. In one (TR), we show that unlearning can be done "for free" because perturbation the parameters is enough to achieve unlearning while keeping the excess risk low enough (Theorem 1). In another (IR), described by our lower bound on the unlearning complexity ratio (Theorem 2), unlearning cannot asymptotically beat retraining through gradient-based methods. In the last regime (ER), in between the other two, our upper bound on the unlearning complexity ratio shows that unlearning is possible at a small fraction of the cost of retraining, a cost that scales with the square of the fraction of forgotten samples (Theorem 3).

Empirical validation confirms these insights, showing the utility of analysing unlearning through the minimax complexity framework. Beyond unveiling fundamental limits and opportunities, our results address an open question on whether unlearning can outperform retraining—and under what circumstances. We introduce the first bounds on the unlearning complexity ratio, as well as the first lower bound on unlearning time.

We hope that the framework and findings presented here will stimulate further studies on machine unlearning in broader contexts, and allow further analysis of a wider class of algorithms and objective functions, as well as data distributions. Specifically, lower-bounding the unlearning complexity ratio for methods beyond the first order, not verifying Assumption 2, or methods relying on the forget set, remains an open challenge.

## Impact Statement

This paper presents work whose goal is to advance the field of Machine Learning and Unlearning. There are many potential societal consequences of our work, none which we feel must be specifically highlighted here.

## Acknowledgements

This work was supported by the French government managed by the Agence Nationale de la Recherche (ANR) through France 2030 program with the references ANR-23-PEIA-005 (REDEEM), ANR-22-FAI1-0003 (TRAIN), ANR-24-IAS2-0001 (Fed-Ops), ANR-19-P3IA-0002 (3IA). This research was also supported in part by the European Network of Excellence dAIEDGE under Grant Agreement Nr. 101120726, by the EU HORIZON MSCA 2023 DN project FINALITY (G.A. 101168816), by the Groupe La Poste, sponsor of the Inria Foundation, in the framework of the FedMalin Inria Challenge.

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

# A. Upper Bounds

*Proof of Theorem 1.* Let $e \geq \frac{r_f}{1-r_f} \left( \frac{r_f}{1-r_f} + \sqrt{d}\kappa_{\epsilon,\delta} \right) e_0$.

Let the unlearning algorithm consist in simply adding Gaussian noise to the previous optimum $\boldsymbol{\theta}^*$. By application of the Gaussian mechanism (Dwork and Roth, 2014), adding i.i.d. Gaussian noise with standard deviation $\kappa_{\epsilon,\delta} \|\boldsymbol{\theta}_r^* - \boldsymbol{\theta}^*\|$ ensure $(\epsilon, \delta)$-Unlearning of the forget set. Using the bound from Lemma C.1, we sample:

$$g \sim \mathcal{N}(0, (\kappa_{\epsilon,\delta} \frac{r_f}{1-r_f} \frac{L}{\mu})^2 I_d), \tag{15}$$

where $I_d$ is the identity matrix in $\mathbb{R}^d$.

Let $\hat{\boldsymbol{\theta}} := \boldsymbol{\theta}^* + g$. We can then bound the expected loss of $\hat{\boldsymbol{\theta}}$:

$$\mathbb{E}\left[\mathcal{L}_r(\hat{\boldsymbol{\theta}}) - \mathcal{L}_r^*\right] \leq \mathbb{E}\left[\mathcal{L}_r(\hat{\boldsymbol{\theta}}) - \mathcal{L}_r(\boldsymbol{\theta}^*)\right] + \mathcal{L}_r(\boldsymbol{\theta}^*) - \mathcal{L}_r^* \tag{16}$$

$$\leq \sqrt{d}\kappa_{\epsilon,\delta} \left(\frac{r_f}{1-r_f}\right)^2 \frac{L^2}{\mu} + \frac{r_f}{1-r_f}\frac{L^2}{\mu} \tag{17}$$

$$\leq r_f \left(r_f + \sqrt{d}\kappa_{\epsilon,\delta}\right) e_0 \tag{18}$$

$$\leq e. \tag{19}$$

$\square$

*Proof of Theorem 3.* According to Lemma C.1, we have

$$\|\boldsymbol{\theta}^* - \boldsymbol{\theta}_r^*\| \leq \frac{r_f}{1-r_f}\frac{L}{\mu} =: R_1. \tag{20}$$

To perform the unlearning, we use the noise + fine-tune method as introduced in Algorithm 2. For the noising part, the standard deviation of the noise that needs to be added to ensure $(\epsilon, \delta)$-Unlearning of $\mathcal{D}_f$ is $\kappa_{\epsilon,\delta} \|\boldsymbol{\theta}^* - \boldsymbol{\theta}_r^*\|$. Thus, we set $\sigma = \kappa_{\epsilon,\delta} R_1$ and define $\widetilde{\boldsymbol{\theta}} := \boldsymbol{\theta}^* + g$, where $g \sim \mathcal{N}(0, \sigma^2)$.

Now, one can notice that

$$\mathbb{E}\left[\left\|\widetilde{\boldsymbol{\theta}} - \boldsymbol{\theta}_r^*\right\|^2\right] = \mathbb{E}\left[\|g\|^2\right] + \|\boldsymbol{\theta}^* - \boldsymbol{\theta}_r^*\|^2 \leq (1 + d\kappa_{\epsilon,\delta}^2)R_1^2. \tag{21}$$

While we do not know the exact distance $\left\|\widetilde{\boldsymbol{\theta}} - \boldsymbol{\theta}_r^*\right\|$ our SGD will need to cover, we have its expectation. Thus, we set the learning rate $\gamma$ to be optimal for a the expectation of the distance, *i.e.,* : $\gamma = \sqrt{\frac{(1+d\kappa_{\epsilon,\delta}^2)R_1^2}{T^U L^2}}$.

Let $\mathcal{A}_\gamma$ be the SSD algorithm with learning rate $\gamma$, as defined in Section 3 of Garrigos and Gower (2023). Using Theorem 9.7 from Garrigos and Gower (2023), we get

$$\mathbb{E}\left[\mathcal{L}_r(\mathcal{A}_\gamma(\widetilde{\boldsymbol{\theta}}, \mathcal{D}_r, T)) - \mathcal{L}_r^*\right] \leq \mathbb{E}\left[\frac{\left\|\widetilde{\boldsymbol{\theta}} - \boldsymbol{\theta}_r^*\right\|^2}{2\gamma T} + \frac{\gamma L^2}{2}\right] \tag{22}$$

$$\leq \frac{(1 + d\kappa_{\epsilon,\delta}^2)R_1^2}{2\gamma T} + \frac{\gamma L^2}{2} \tag{23}$$

$$\leq \frac{LR_1}{\sqrt{T}}\sqrt{1 + d\kappa_{\epsilon,\delta}^2}. \tag{24}$$

For a given excess risk threshold $e$, the unlearning time can then be upper-bounded as

$$T_e^U \leq \frac{L^2 R_1^2}{e^2}(1 + d\kappa_{\epsilon,\delta}^2) = \left(\frac{r_f}{1 - r_f}\right)^2 (1 + d\kappa_{\epsilon,\delta}^2) \left(\frac{e_0}{e}\right)^2 . \tag{25}$$

$\square$

## B. Lower bounds

The lower bounds of Section 4 rely on three steps: 1) defining a class of objective functions $\mathcal{L}^g$ for $g : \mathbb{R}^s \to \{-1, 1\}$ such that their optimum over $\mathcal{D}$ does not provide any information on the dataset $\mathcal{D}_r$, 2) showing that two such functions $\mathcal{L}^g$ and $\mathcal{L}^{-g}$ have optimums over $\mathcal{D}_r$ distant from one another, and 3) showing that any algorithm's output will behave nearly identically on both $\mathcal{L}^g$ and $\mathcal{L}^{-g}$, thus leading to the impossibility of having both functions efficiently optimized by the same algorithm.

In what follows, for any function $g : \mathbb{R}^s \to [-1, 1]$, we denote as $\mathcal{L}^g(\boldsymbol{\theta}) = \mathbb{E}\left[\ell^g(\boldsymbol{\theta}, \xi)\right]$ where $\ell^g$ is a loss function such that

$$\ell^g(\boldsymbol{\theta}, \xi) = \frac{\mu}{2}\|\boldsymbol{\theta}\|^2 - \frac{L}{2}g(\xi)\boldsymbol{\theta}_1 , \tag{26}$$

where $\boldsymbol{\theta}_1$ is the first coordinate of $\boldsymbol{\theta}$ in the canonical basis of $\mathbb{R}^d$.

This loss was used as a base for the experiment on synthetic data. More specifically, we added an $L1$ penalization term in order to make the function non-smooth, and expanded the loss on $\theta_1$ to the first half of parameters. This is motivated by the need to avoid numerical instability as every other parameter but the first would otherwise quickly converge to 0, and the first one would oscillate around the optimal value, in a process that could converge prematurely.

By definition, $\nabla_{\boldsymbol{\theta}}\ell^g(\boldsymbol{\theta}, \xi) = \mu\boldsymbol{\theta} - Lg(\xi)e_1/2$ where $e_1$ is the first vector of the canonical basis of $\mathbb{R}^d$, and $\ell^g$ is $L$-Lipschitz and $\mu$-strongly convex. Moreover, the objective function on $\mathcal{D}_r$ is $\mathcal{L}_r^g(\boldsymbol{\theta}) = \frac{\mu}{2}\|\boldsymbol{\theta}\|^2 - \frac{L}{2}\mathbb{E}\left[g(\xi')\right]\boldsymbol{\theta}_1$, where $\xi' \sim \mathcal{D}_r$, and thus the minimizer of $\mathcal{L}_r^g$ is $\boldsymbol{\theta}_{g,r}^* = \frac{L}{2\mu}\mathbb{E}\left[g(\xi')\right]e_1$ (note that $\|\boldsymbol{\theta}_{g,r}^*\| = \frac{L}{2\mu}|\mathbb{E}\left[g(\xi')\right]| \leq R$). We now show that, provided we find two functions $g, g'$ such that the output of any algorithm is (statistically) almost indistinguishable, then minimizing both $\mathcal{L}^g$ and $\mathcal{L}^{g'}$ beyond a certain quantity is impossible. To properly define this *indistinguishability*, we will use the *total variation* distance $d_{\text{TV}}(P, Q) = \sup_{A \subset \mathbb{R}^s} |P(A) - Q(A)|$ for two probability distributions $P$ and $Q$.

**Lemma B.1.** *Let $g, g' : \mathbb{R}^s \to [-1, 1]$ two functions, $\theta_0, \theta_0' \in \mathbb{R}^d$ two initial parameters, and $A \in \boldsymbol{A}$ an algorithm. Then*

$$\sup_{g'' \in \{g, g'\}} \mathbb{E}\left[\mathcal{L}_r^{g''}(\boldsymbol{\theta}_T^A(\boldsymbol{\theta}_0, \ell^{g''}, \mathcal{D}_r)) - \mathcal{L}_r^{g''*}\right] \geq \frac{L^2(\mathbb{E}\left[g(\xi')\right] - \mathbb{E}\left[g'(\xi')\right])^2}{32\,\mu}(1 - d_{\text{TV}}(P_g, P_{g'})), \tag{27}$$

*where $\xi' \sim \mathcal{D}_r$, $\mathcal{L}_r^{g''*} = \min_{\boldsymbol{\theta} \in \mathbb{R}^d} \mathcal{L}_r^{g''}(\boldsymbol{\theta})$ and $P_g$ (resp. $P_{g'}$) is the probability distribution of $\boldsymbol{\theta}_T^A(\boldsymbol{\theta}_0, \ell^g, \mathcal{D}_r)$ (resp. $\boldsymbol{\theta}_T^A(\theta_0', \ell^{g'}, \mathcal{D}_r)$).*

*Proof.* First, note that $\boldsymbol{\theta}_{g,r}^* = \frac{L}{2\mu}\mathbb{E}\left[g(\xi')\right]e_1$ and thus $\mathcal{L}_r^g(\boldsymbol{\theta}) - \mathcal{L}_r^{g*} = \frac{\mu}{2}\|\boldsymbol{\theta}_{g,r}^* - \boldsymbol{\theta}\|^2$. Using the optimal transport definition of total variation (see *e.g.,* Villani et al. 2009), $d_{\text{TV}}(P, Q) = \inf_{(X,Y)} \mathbb{P}(X \neq Y)$ where the infimum is taken over all couplings of $P$ and $Q$. As a consequence, there exists two random variables $\boldsymbol{\theta}_1 \sim P_g$ and $\boldsymbol{\theta}_2 \sim P_{g'}$, and such that $\mathbb{P}(\boldsymbol{\theta}_1 \neq \boldsymbol{\theta}_2) = d_{\text{TV}}(P_g, P_{g'})$, leading to

$$
\begin{aligned}
\sup_{g'' \in \{g, g'\}} \mathbb{E}\left[\left\|\boldsymbol{\theta}_{g'',r}^* - \boldsymbol{\theta}_T^A(\boldsymbol{\theta}_0, \ell^{g''}, \mathcal{D}_r)\right\|^2\right] &= \max\left\{\mathbb{E}\left[\left\|\boldsymbol{\theta}_{g,r}^* - \boldsymbol{\theta}_1\right\|^2\right], \mathbb{E}\left[\left\|\boldsymbol{\theta}_{g',r}^* - \boldsymbol{\theta}_2\right\|^2\right]\right\} \\
&\geq (1 - d_{\text{TV}}(P_g, P_{g'})) \max\left\{\left\|\boldsymbol{\theta}_{g,r}^* - \tilde{\boldsymbol{\theta}}\right\|^2, \left\|\boldsymbol{\theta}_{g',r}^* - \tilde{\boldsymbol{\theta}}\right\|^2\right\} \\
&\geq (1 - d_{\text{TV}}(P_g, P_{g'}))\frac{\|\boldsymbol{\theta}_{g,r}^* - \boldsymbol{\theta}_{g',r}^*\|^2}{4} ,
\end{aligned} \tag{28}
$$

where $\tilde{\boldsymbol{\theta}} = \mathbb{E}\left[\boldsymbol{\theta}_1 \mid \boldsymbol{\theta}_1 = \boldsymbol{\theta}_2\right] = \mathbb{E}\left[\boldsymbol{\theta}_2 \mid \boldsymbol{\theta}_1 = \boldsymbol{\theta}_2\right]$. Finally, using the formula for $\boldsymbol{\theta}_{g,r}^* = \frac{L}{2\mu}\mathbb{E}\left[g(\xi')\right]e_1$ and $\mathcal{L}_r^g(\boldsymbol{\theta}) - \mathcal{L}_r^{g*} = \frac{\mu}{2}\|\boldsymbol{\theta}_{g,r}^* - \boldsymbol{\theta}\|^2$ gives the desired result. $\square$

We now show that a particular choice of functions $g, g'$ leads to almost indistinguishable outputs.

**Lemma B.2.** *Assume that* $\forall \gamma \in [0, 1]$, *there exists* $A^\gamma \subset supp(\mathcal{D}_r)$ *such that* $\mathbb{P}_{\mathcal{D}_r}(A) = (1 + \gamma)/2$. *Let* $g^\gamma(\xi) = 2\mathbf{1}\{\xi \in A^\gamma\} - 1$ *if* $\xi \in supp(\mathcal{D}_r)$, *and* $g^\gamma(\xi) = -\min\{1, \frac{(1-r_f)\gamma}{r_f}\}$ *otherwise. Then, we have*

$$d_{\mathrm{TV}}\left(\mathcal{U}(T, \ell^{g^\gamma}, \mathcal{D}_r, \mathcal{D}_f), \mathcal{U}(T, \ell^{g^0}, \mathcal{D}_r, \mathcal{D}_f)\right) \leq \frac{\pi\gamma\sqrt{T}}{4} + \frac{((1-r_f)\gamma - r_f)_+}{2r_f}(e^\epsilon - 1 + \delta). \tag{29}$$

*Proof.* First, note that the minimizer of $\mathcal{L}^\gamma$ is $\boldsymbol{\theta}^*_\gamma = \frac{L}{2\mu}((1 - r_f)\gamma - r_f)_+ e_1$, and we thus have $\mathcal{U}(T, \ell^{g^\gamma}, \mathcal{D}_r, \mathcal{D}_f) = \boldsymbol{\theta}^A_T(\boldsymbol{\theta}^*_\gamma, \ell^{g^\gamma}, \mathcal{D}_r)$ and $\mathcal{U}(T, \ell^{-g^\gamma}, \mathcal{D}_r, \mathcal{D}_f) = \boldsymbol{\theta}^A_T(-\boldsymbol{\theta}^*_\gamma, \ell^{-g^\gamma}, \mathcal{D}_r)$. To ease the notations, we denote by $\boldsymbol{\theta}_{k,l} = \boldsymbol{\theta}^A_T(\frac{(1-r_f)L\gamma_{2k+1}}{2\mu}e_1, \ell^{g^{\gamma_{2l}}}, \mathcal{D}_r)$ the output of algorithm $A$ on the function $\ell^{g^{\gamma_{2l}}}$ starting at $\boldsymbol{\theta}_0 = \frac{(1-r_f)L\gamma_{2k+1}}{2\mu}e_1$, where $\gamma_k = \left(\gamma - \frac{kr_f}{1-r_f}\right)_+$. Let $K = \left\lceil\frac{(1-r_f)\gamma}{2r_f}\right\rceil$, then we have, by triangular inequality,

$$d_{\mathrm{TV}}(\boldsymbol{\theta}_{0,0}, \boldsymbol{\theta}_{K,K}) \leq \sum_{k=0}^{K-1} d_{\mathrm{TV}}(\boldsymbol{\theta}_{k,k}, \boldsymbol{\theta}_{k,k+1}) + \sum_{k=0}^{K-1} d_{\mathrm{TV}}(\boldsymbol{\theta}_{k,k+1}, \boldsymbol{\theta}_{k+1,k+1}). \tag{30}$$

By construction, we have $\boldsymbol{\theta}_{0,0} = \mathcal{U}(T, \ell^{g^\gamma}, \mathcal{D}_r, \mathcal{D}_f)$ and $\boldsymbol{\theta}_{K,K} = \mathcal{U}(T, \ell^{g^0}, \mathcal{D}_r, \mathcal{D}_f)$. We now show that both sums can be bounded: the first using the fact that the $T$ gradients $g^\gamma(\xi_t)$ for $t \in [\![0, T - 1]\!]$ are close in total variation distance (*i.e.,* Lemma C.5), and the second using the $(\epsilon, \delta)$-Unlearning constraint on $\mathcal{U}$.

By Lemma C.4, there is a measurable function $\varphi_A$ such that $\boldsymbol{\theta}_{k,l} = \varphi_A\left(\frac{(1-r_f)L\gamma_{2k+1}}{2\mu}, Z_0^l, \ldots, Z_{T-1}^l, \omega\right)$ where $Z_t^l = (1 + g^{\gamma_{2l}}(\xi_t))/2$ are i.i.d. Bernoulli random variables of parameter $\frac{1+\gamma_{2l}}{2}$. As $\boldsymbol{\theta}_{k,k}$ and $\boldsymbol{\theta}_{k,k+1}$ are outputs of the same algorithm initialized at the same starting position, we have

$$\begin{aligned}
\sum_{k=0}^{K-1} d_{\mathrm{TV}}(\boldsymbol{\theta}_{k,k}, \boldsymbol{\theta}_{k,k+1}) &= \sum_{k=0}^{K-1} d_{\mathrm{TV}}\left((Z_0^k, \ldots, Z_{T-1}^k), (Z_0^{k+1}, \ldots, Z_{T-1}^{k+1})\right) \\
&= d_{\mathrm{TV}}\left(\mathrm{Bin}\left(T, \frac{1+\gamma_{2k}}{2}\right), \mathrm{Bin}\left(T, \frac{1+\gamma_{2k+2}}{2}\right)\right) \\
&= \sum_{k=0}^{K-1} \frac{\sqrt{T}}{2}\left|\tan^{-1}\left(\frac{\gamma_{2k+2}}{\sqrt{1-\gamma_{2k+2}^2}}\right) - \tan^{-1}\left(\frac{\gamma_{2k}}{\sqrt{1-\gamma_{2k}^2}}\right)\right| \\
&= \frac{\sqrt{T}}{2}\left|\tan^{-1}\left(\frac{\gamma_{2K}}{\sqrt{1-\gamma_{2K}^2}}\right) - \tan^{-1}\left(\frac{\gamma_0}{\sqrt{1-\gamma_0^2}}\right)\right| \\
&= \frac{\sqrt{T}}{2}\tan^{-1}\left(\frac{\gamma_{2K}}{\sqrt{1-\gamma_{2K}^2}}\right) \\
&\leq \frac{\pi\gamma\sqrt{T}}{4},
\end{aligned} \tag{31}$$

using the fact that $f : x \mapsto \tan^{-1}\left(\frac{x}{\sqrt{1-x^2}}\right)$ is increasing and convex on $x \in [0, 1]$, and $f(0) = 0$ and $f(1) = \pi/2$.

Finally, let $\mathcal{D}'_f$ be a probability distribution on $\mathbb{R}^s$ such that $supp(\mathcal{D}'_f) \cap (supp(\mathcal{D}_r) \cup supp(\mathcal{D}_f)) = \emptyset$, for any $\gamma \in [0, 1]$, let $\tilde{g}^\gamma(\xi) = g^\gamma(\xi)$ if $\xi \in supp(\mathcal{D}_r) \cup supp(\mathcal{D}_f)$, and $\tilde{g}^\gamma(\xi) = 1$ otherwise. Then, we have $\mathcal{U}(T, \ell^{\tilde{g}^{\gamma_{2k}}}, \mathcal{D}_r, \mathcal{D}_f) = \boldsymbol{\theta}_{k,k}$ and $\mathcal{U}(T, \ell^{\tilde{g}^{\gamma_{2k}}}, \mathcal{D}_r, \mathcal{D}'_f) = \boldsymbol{\theta}_{k-1,k}$. Thus, we have

$$\begin{aligned}
\sum_{k=0}^{K-1} d_{\mathrm{TV}}(\boldsymbol{\theta}_{k,k+1}, \boldsymbol{\theta}_{k+1,k+1}) &= \sum_{k=0}^{K'-1} d_{\mathrm{TV}}\left(\mathcal{U}(T, \ell^{\tilde{g}^{\gamma_{2k+2}}}, \mathcal{D}_r, \mathcal{D}_f), \mathcal{U}(T, \ell^{\tilde{g}^{\gamma_{2k+2}}}, \mathcal{D}_r, \mathcal{D}'_f)\right) \\
&\leq \sum_{k=0}^{K'-1}(e^\epsilon - 1 + \delta) \\
&= K'(e^\epsilon - 1 + \delta),
\end{aligned} \tag{32}$$

where $K' = \left\lceil\frac{((1-r_f)\gamma - r_f)_+}{2r_f}\right\rceil$. Combining the two inequalities concludes the proof. $\square$

We are now in position to prove Theorem 2.

*Proof of Theorem 2.* Combining Lemma B.1 (with $g = g^\gamma$ and $g' = -g^\gamma$) and Lemma B.2, we have, for any $\gamma \in [0, 1]$,

$$\min_{\mathcal{U}\in\mathbb{U}} \max_{\mathcal{L}\in\mathcal{F}_{sc}} \mathbb{E}\left[\mathcal{L}_r(\mathcal{U}(T, \ell, \mathcal{D}_r, \mathcal{D}_f)) - \mathcal{L}_r^*\right] \geq \frac{L^2\gamma^2}{8\mu}(1 - d_{\mathrm{TV}}(P_{g^\gamma}, P_{-g^\gamma})), \tag{33}$$

where $d_{\text{TV}}(P_{g^\gamma}, P_{-g^\gamma}) \leq d_{\text{TV}}(P_{g^\gamma}, P_{g^0}) + d_{\text{TV}}(P_{g^0}, P_{-g^\gamma}) \leq \frac{\pi\gamma\sqrt{T}}{2} + \frac{((1-r_f)\gamma - r_f)_+}{r_f}(e^\epsilon - 1 + \delta)$. Let $c_1, c_2 \in [0, 1]$ and $\gamma = c_1/\sqrt{T}$. If $\left(\frac{(1-r_f)c_1}{r_f\sqrt{T}} - 1\right)_+ (e^\epsilon - 1 + \delta) \leq c_2$, then

$$\min_{\mathcal{U} \in \mathbb{U}} \max_{\mathcal{L} \in \mathcal{F}_{sc}} \mathbb{E}\left[\mathcal{L}_r(\mathcal{U}(T, \ell, \mathcal{D}_r, \mathcal{D}_f)) - \mathcal{L}_r^*\right] \geq \frac{L^2 c_1^2}{8\,\mu T}\left(1 - \frac{\pi c_1}{2} - c_2\right), \tag{34}$$

and thus, if $\left(\frac{(1-r_f)\sqrt{8\mu e}}{r_f L\sqrt{1 - \frac{\pi c_1}{2} - c_2}} - 1\right)_+ (e^\epsilon - 1 + \delta) \leq c_2$,

$$T_e^U \geq \frac{L^2 c_1^2}{8\,\mu e}\left(1 - \frac{\pi c_1}{2} - c_2\right). \tag{35}$$

Finally, we take $c_2 = 1/2$, $c_1$ such that $1 - \frac{\pi c_1}{2} - c_2 = 1/3$, and rewrite the condition as $e \leq \frac{r_f^2 L^2}{8(1-r_f)^2\mu}(1 - \frac{\pi c_1}{2} - c_2)\left(1 + \frac{c_2}{e^\epsilon - 1 + \delta}\right)^2$. A simple functional analysis gives that, for $10^{-8} \leq \delta \leq \epsilon$, we have

$$1 + \frac{1}{2(e^\epsilon - 1 + \delta)} \geq 1 + \frac{1}{2(e^\epsilon - 1 + \epsilon)} \geq c_3\left(1 + \frac{\sqrt{2\ln(1.25 \cdot 10^8)}}{\epsilon}\right) \geq c_3\left(1 + \frac{\sqrt{2\ln(1.25/\delta)}}{\epsilon}\right), \tag{36}$$

where $c_3 = 1/\sqrt{32\ln(1.25 \cdot 10^8)}$ and the desired result. $\qquad\qquad\square$

Using the same approach, a lower bound on the time complexity of scratch can also be derived.

*Proof of Lemma 4.2.* First, by strong convexity, we have

$$\mathcal{L}_r(0) - \mathcal{L}_r^* \leq \langle \nabla\mathcal{L}(0), \boldsymbol{\theta}_r^* \rangle - \frac{\mu}{2}\|\boldsymbol{\theta}_r^*\|^2. \tag{37}$$

Moreover, the convexity of $\boldsymbol{\theta} \mapsto \mathcal{L}_r(\boldsymbol{\theta}) - \frac{\mu}{2}\|\boldsymbol{\theta}\|^2$ implies that $\langle \nabla\mathcal{L}(-\boldsymbol{\theta}_r^*) + \mu\boldsymbol{\theta}_r^* - \nabla\mathcal{L}(0), -\boldsymbol{\theta}_r^* \rangle \geq 0$ and thus

$$\mathcal{L}_r(0) - \mathcal{L}_r^* \leq \|\nabla\mathcal{L}(-\boldsymbol{\theta}_r^*)\|\,\|\boldsymbol{\theta}_r^*\| - \frac{3\mu}{2}\|\boldsymbol{\theta}_r^*\|^2 \leq LR - \frac{3\mu}{2}R^2 = \frac{L^2}{8\mu}. \tag{38}$$

As a consequence, if $e \geq e_0$, then $T_e^S = 0$ (and $T_e^U = 0$). Let us now assume that $e < e_0$. First, note that this convergence rate is achieved by stochastic gradient descent. For example, a direct extension of Theorem 6.2 from Bubeck et al. (2015) gives, after $T$ iterations of (stochastic) gradient descent $\theta_{t+1} = \theta_t - \eta_t\nabla_\theta\ell(\theta_t, \xi_t)$ with decreasing step-size $\eta_t = \frac{2}{\mu(t+2)}$.

$$\mathbb{E}\left[\mathcal{L}\left(\tilde{\theta}_T\right) - \min_{\theta \in \mathbb{R}^d}\mathcal{L}(\theta)\right] \leq \frac{2L^2}{\mu(T+2)}, \tag{39}$$

where $\tilde{\theta}_T = \sum_{t=0}^{T-1}\frac{2(1+t)}{(T+1)(T+2)}\theta_t$, and thus

$$T_e^S \leq \frac{2L^2}{\mu e}. \tag{40}$$

The lower bound is a consequence of Theorem 2 with $\kappa_{\epsilon,\delta} = 0$, as an algorithm retraining from scratch would not depend on the forget dataset, and thus have absolute privacy. In particular, if $e \leq \frac{L^2}{8\mu}(1 - \frac{\pi c_1}{2})$, we have

$$T_e^S \geq \frac{L^2 c_1^2}{8\,\mu e}(1 - \frac{\pi c_1}{2}), \tag{41}$$

and as soon as $e/e_0 \leq 1 - \eta$ for $\eta > 0$, there exists a constant $c_2 > 0$ such that $T_e^S \geq c_2 e_0/e$. $\qquad\square$

## C. Useful lemmas

In this section, we provide five lemmas that will be necessary to prove our upper and lower bounds (see sections above), as well as the proof for unlearning definition equivalence.

**Lemma C.1.** *Let* $\boldsymbol{\theta}_r^* = \arg\min_{\boldsymbol{\theta}} \mathcal{L}_r(\boldsymbol{\theta})$. *Then, we have:*

$$\|\boldsymbol{\theta}^* - \boldsymbol{\theta}_r^*\| \leq \frac{r_f}{1 - r_f} \cdot \frac{L}{\mu} . \tag{42}$$

*Proof.* By strong convexity of $\mathcal{L}_r$, we have $\|\boldsymbol{\theta}^* - \boldsymbol{\theta}_r^*\| \leq \frac{\|\nabla \mathcal{L}_r(\boldsymbol{\theta}^*)\|}{\mu}$. Moreover, $\|\nabla \mathcal{L}_r(\boldsymbol{\theta}^*)\| = \|\mathbb{E}\left[\nabla \ell(\boldsymbol{\theta}^*, \xi_r)\right]\| = \left\|-\frac{r_f}{1-r_f} \mathbb{E}\left[\nabla \ell(\boldsymbol{\theta}^*, \xi_r)\right]\right\| \leq \frac{r_f}{1-r_f} L$ where $\xi_r \sim \mathcal{D}_r$ and $\xi_f \sim \mathcal{D}_f$, as $\nabla \mathcal{L}(\boldsymbol{\theta}^*) = 0$. Combining the two inequalities gives the desired result. $\square$

**Lemma C.2.**

$$\mathcal{L}_r(\boldsymbol{\theta}^*) - \mathcal{L}_r^* \leq \left(\frac{r_f}{1-r_f}\right)^2 \frac{L^2}{\mu} \tag{43}$$

*Proof.* Let $\boldsymbol{\theta}_r^* = \arg\min_{\boldsymbol{\theta}} \mathcal{L}_r(\boldsymbol{\theta})$. Then,

$$\mathbb{E}_{\mathcal{D}_r}\left(\ell(\boldsymbol{\theta}^*) - \ell(\boldsymbol{\theta}_r^*)\right) = \mathbb{E}_{\mathcal{D}}\left(\ell(\boldsymbol{\theta}^*) - \ell(\boldsymbol{\theta}_r^*)\right) - \frac{r_f}{1-r_f}\mathbb{E}_{\mathcal{D}_f}\left(\ell(\boldsymbol{\theta}^*) - \ell(\boldsymbol{\theta}_r^*)\right) \tag{44}$$

$$\leq -\frac{r_f}{1-r_f}\mathbb{E}_{\mathcal{D}_f}\left(\ell(\boldsymbol{\theta}^*) - \ell(\boldsymbol{\theta}_r^*)\right) \tag{45}$$

$$\leq \frac{r_f}{1-r_f}L\|\boldsymbol{\theta}^* - \boldsymbol{\theta}_r^*\| \tag{46}$$

$$\leq \left(\frac{r_f}{1-r_f}\right)^2 \frac{L^2}{\mu}, \tag{47}$$

where the last inequality is given by Lemma C.1. $\square$

**Lemma C.3.** *If the unlearning algorithm $\mathcal{U}$ verifies $(\epsilon, \delta)$-Unlearning, then, for any triplet of distributions $(\mathcal{D}_r, \mathcal{D}_f, \mathcal{D}_f')$, we have*

$$d_{\mathrm{TV}}\left(\mathcal{U}(\mathcal{D}_r, \mathcal{D}_f), \mathcal{U}(\mathcal{D}_r, \mathcal{D}_f')\right) \leq e^\epsilon - 1 + \delta . \tag{48}$$

*Proof.* By $(\epsilon, \delta)$-Unlearning, we have, for any $S \subset \mathbb{R}^s$, $\mathbb{P}[\mathcal{U}(\mathcal{D}_r, \mathcal{D}_f) \in S] \leq e^\epsilon \cdot \mathbb{P}[\mathcal{U}(\mathcal{D}_r, \mathcal{D}_f') \in S] + \delta$, and thus

$$\mathbb{P}[\mathcal{U}(\mathcal{D}_r, \mathcal{D}_f) \in S] - \mathbb{P}[\mathcal{U}(\mathcal{D}_r, \mathcal{D}_f') \in S] \leq (e^\epsilon - 1) \cdot \mathbb{P}[\mathcal{U}(\mathcal{D}_r, \mathcal{D}_f') \in S] + \delta \leq e^\epsilon - 1 + \delta . \tag{49}$$

The converse relation with $\mathcal{D}_f$ and $\mathcal{D}_f'$ exchanged leads to a bound on the absolute value, and thus the desired result. $\square$

*Proof of Lemma 3.1.* Let $\mathcal{D}_0$ be an arbitrary distribution, *e.g.,* the uniform distribution on the $R/2$-ball. Let $\mathcal{U} \in \mathbb{U}$ be an $(\epsilon, \delta)$-Unlearning algorithm. Let $\mathcal{U} \in \mathbb{U}$ be an $(\epsilon, \delta)$-Unlearning algorithm. Then, the algorithm $\mathcal{A}_0 : (T, l, \mathcal{D}_r) \longmapsto \mathcal{U}(T, l, \mathcal{D}_r, \mathcal{D}_0)$ is such that for any couple of distributions $(\mathcal{D}_r, \mathcal{D}_f)$ over $\mathbb{B}(0, R)$ and subset $S \subset \mathbb{R}^d$,

$$\mathbb{P}[\mathcal{U}(\mathcal{D}_r, \mathcal{D}_f) \in S] \leq e^\epsilon \cdot \mathbb{P}[\mathcal{U}(\mathcal{D}_r, \mathcal{D}_0) \in S] + \delta = e^\epsilon \cdot \mathbb{P}[\mathcal{A}_0(\mathcal{D}_r) \in S] + \delta,$$
$$\mathbb{P}[\mathcal{A}_0(\mathcal{D}_r) \in S] = \mathbb{P}[\mathcal{U}(\mathcal{D}_r, \mathcal{D}_0) \in S] \leq e^\epsilon \cdot \mathbb{P}[\mathcal{U}(\mathcal{D}_r, \mathcal{D}_f) \in S] + \delta.$$

$\mathcal{U}$ is thus an $(\epsilon, \delta)$-Reference Unlearning algorithm. This proves the first implication.

Let $\mathcal{U} \in \mathbb{U}$ be an $(\epsilon, \delta)$-Reference Unlearning algorithm and $\mathcal{A} \in \mathbb{A}$ its reference algorithm. Let $\mathcal{D}_r, \mathcal{D}_f, \mathcal{D}_f'$ be three distributions over $\mathbb{B}(0, R)$. Then,

$$\mathbb{P}[\mathcal{U}(\mathcal{D}_r, \mathcal{D}_f) \in S] \leq e^\epsilon \cdot \mathbb{P}[\mathcal{A}(\mathcal{D}_r) \in S] + \delta \leq e^\epsilon\left(e^\epsilon \cdot \mathbb{P}[\mathcal{U}(\mathcal{D}_r, \mathcal{D}_f') \in S] + \delta\right) + \delta \tag{50}$$

$\mathcal{U}$ is thus an $(2\epsilon, (1 + \exp(\epsilon))\delta)$-Reference Unlearning algorithm. This concludes the proof. $\square$

**Lemma C.4.** *Let $A \in \mathbf{A}$ be an iterative algorithm as defined in Algorithm 1. Then, there exists $T$ i.i.d. random variables $\xi_t \sim \mathcal{D}_r$ and a measurable function $\varphi_{\mathcal{A}}$ such that, for any $T > 0$, $\theta_0 \in \mathbb{R}^d$ and $g : \mathbb{R}^s \to \{-1, 1\}$, we have*

$$\theta_T^A(\theta_0, \ell^g, \mathcal{D}_r) = \varphi_A(\theta_0, g(\xi_0), \ldots, g(\xi_{T-1}), \omega). \tag{51}$$

*Proof.* Let $(m_t, \theta_t)$ be the memory state and current parameter of algorithm $A$ at iteration $t$ (see algorithm 1). First, for $T = 0$, $m_0 = \emptyset$ and $\theta_0$ are both (trivially) measurable functions of $\theta_0$. Then, by recursion, if both $m_{T-1}$ and $\theta_{T-1}$ are measurable functions of $\theta_0, g(\xi_0), \ldots, g(\xi_{T-2}), \omega$, then

$$(m_T, \theta_T) = A(m_t, \nabla \ell^g(\theta_{T-1}, \xi_{T-1}), \omega) = A(m_{T-1}, \mu \theta_{T-1} - Lg(\xi_{T-1})e_1/2, \omega), \tag{52}$$

which is a measurable function of $\theta_0, g(\xi_0), \ldots, g(\xi_{T-1}), \omega$. This concludes the proof. $\square$

**Lemma C.5.** *Let $T \geq 0$ and $\gamma, \gamma' \in [-1, 1]$. Then, we have*

$$d_{\mathrm{TV}}\left(\mathit{Bin}\left(T, \frac{1+\gamma}{2}\right), \mathit{Bin}\left(T, \frac{1+\gamma'}{2}\right)\right) \leq \frac{\sqrt{T}}{2}\left|\tan^{-1}\left(\frac{\gamma'}{\sqrt{1-\gamma'^2}}\right) - \tan^{-1}\left(\frac{\gamma}{\sqrt{1-\gamma^2}}\right)\right|. \tag{53}$$

*Proof.* Assume that $\gamma' \geq \gamma$, and let $\varphi(\gamma, \gamma') = d_{\mathrm{TV}}\left(\mathrm{Bin}\left(T, \frac{1+\gamma}{2}\right), \mathrm{Bin}\left(T, \frac{1+\gamma'}{2}\right)\right)$. The proof relies on bounding the derivative of $\varphi$ with respect to its second variable. Let $\gamma' = \gamma + \varepsilon$ where $\varepsilon > 0$, then

$$
\begin{aligned}
\varphi(\gamma, \gamma + \varepsilon) &= \frac{1}{2}\mathbb{E}\left[\left|1 - \frac{P_{\gamma+\varepsilon}(X)}{P_\gamma(X)}\right|\right] \\
&= \frac{1}{2}\mathbb{E}\left[\left|1 - \frac{(1+\gamma+\varepsilon)^X(1-\gamma-\varepsilon)^{T-X}}{(1+\gamma)^X(1-\gamma)^{T-X}}\right|\right] \\
&= \frac{1}{2}\mathbb{E}\left[\left|1 - (1 + X\frac{\varepsilon}{1+\gamma})(1 - (T-X)\frac{\varepsilon}{1-\gamma}) + O(\varepsilon^2)\right|\right] \\
&= \frac{\varepsilon}{2(1+\gamma)}\mathbb{E}\left[\left|X - (T-X)\frac{1+\gamma}{1-\gamma}\right|\right] + O(\varepsilon^2) \\
&\leq \frac{\varepsilon}{(1+\gamma)}\sqrt{\frac{\mathrm{Var}(X)}{(1-\gamma)^2}} + O(\varepsilon^2) \\
&= \frac{\varepsilon}{2}\sqrt{\frac{T}{1-\gamma^2}} + O(\varepsilon^2),
\end{aligned}
\tag{54}
$$

where $P_\gamma$ is the density of the binomial distribution $\mathrm{Bin}\left(T, \frac{1+\gamma}{2}\right)$, and $X \sim \mathrm{Bin}\left(T, \frac{1+\gamma}{2}\right)$. As the total variation distance verifies the triangular inequality, we have

$$d_{\mathrm{TV}}\left(\mathrm{Bin}\left(T, \frac{1+\gamma}{2}\right), \mathrm{Bin}\left(T, \frac{1+\gamma'}{2}\right)\right) \leq \int_{u=\gamma}^{\gamma'} \varphi(u, u+du) \leq \frac{\sqrt{T}}{2}\left(\tan^{-1}\left(\frac{\gamma'}{\sqrt{1-\gamma'^2}}\right) - \tan^{-1}\left(\frac{\gamma}{\sqrt{1-\gamma^2}}\right)\right).$$

$$\tag{55}$$
$\square$

## D. Experimental Details

In order to produce Fig. 2, we optimized a standard cross-entropy loss with $L2$ regularization (of weight 1). We used a batch-size of 64 and trained until the threshold $e$ was reached, for every chosen value of $\kappa_{\epsilon,\delta}$. The 60 values of $e$ (resp. $\kappa_{\epsilon,\delta}$) are chosen regularly spaced on the linear scale between $5 * 10^{-6}$ and 10 (resp. $10^{-2}$ and $10^2$). The optimized used is the standard SGD optimizer without acceleration. The learning rate is initialised at $10^{-2}$ and multiplied by 0.6 every 1000 epoch. Each experiment is repeated 50 times and the results are then averaged.

