# OpenReview forum: "When to Forget? Complexity Trade-offs in Machine Unlearning"
_ICML.cc/2025/Conference — ICML 2025 poster_

### Official Review · Reviewer_tKga · 2025-03-04

**Overall Recommendation:** 3

**Summary:**

The paper considers machine unlearning for strongly convex objective functions and under the assumption that the forget set is inaccessible to the unlearning method. The paper provides and proves some theoretical lower and upper bounds on the ratio of the number of training iterations needed by unlearning from the fully trained model and retraining from the initial model to achieve a given excess risk threshold. The bound depends on the weight of the forget set and data dimensionality. The paper identifies three regimes where 1) trivial unlearning is possible by adding noise to the parameters, 2) unlearning is inefficient (impossible) and does not asymptotically improve the number of iterations and 3) efficient unlearning is possible by adding noise before fine tuning with stochastic gradient descent.

## update after rebuttal
The authors have satisfactorily addressed my concerns and my review was already positive.

**Claims And Evidence:**

The claims made are supported by clear and convincing evidence. However, there are some minor issues
1) The paper makes the strong assumption of strongly convex and Lipschitz smooth loss functions. While this seems standard in previous work, their applicability in practice needs to be clarified.
2) Definitions 1 and 2 are slightly non-standard and strong as they are for _any_ forget sets. In the usual DP definitions, there is some constraints regarding neighbouring forget sets.
3) In Section 4.3, inefficient unlearning seems to be a more appropriate description than impossible unlearning. Unlearning is possible and may even more efficient but the speedup is not an asymptotic order.

**Essential References Not Discussed:**

Essential references (Chourasia and Shah, 2023; Allouah et al., 2024) are discussed.

**Experimental Designs Or Analyses:**

There is only one experiment on linear regression model on synthetic function and dataset instead of real world datasets such as the Californian Housing dataset.
I am not sure why the expected loss function in line 389 is used over other loss functions.

**Methods And Evaluation Criteria:**

Not applicable, this is a theory work.

**Other Comments Or Suggestions:**

Typos
*  In Sec 4, there should a brief summary about how each theorem is proven. For example, Theorem 1 uses the MU algorithm that adds Gaussian noise to the previous optimum
* "Let the unlearning algorithm consist in simply" in line 554
* Distribution expectation is taken w.r.t. in Lemma C.1 proof is missing

**Other Strengths And Weaknesses:**

**Strengths**
1. The paper and mathematical definitions (especially Section 1-3) are well-written and clear.
2. The paper provides novel and important theoretical insights that would work for any stochastic, iterative and first-order algorithm.

**Weaknesses**
1. The experiments are limited. It is on a synthetic dataset and loss function.

**Questions For Authors:**

1. Allouah et al., 2024 have separate results from out of distribution data. How do out of distribution data affect your theoretical guarantees? The DP-based MU algorithms and definition seem to work for any dataset as long as the weight on the forget set is fixed. The response would make the difference between both papers clearer.
3. Explain assumption 2 and why it is "verified for continuous distribution"
4. Why is it appropriate to use the expected loss function in line 389 over other loss functions? The response would affect the evaluation of the experiments

**Relation To Broader Scientific Literature:**

(Chourasia and Shah, 2023; Allouah et al., 2024) have identified upper-bounds on the number of gradient steps needed by their specific MU algorithm and retraining to achieve unlearning.
The paper identified lower and upper bounds of the ratio of the number of training iterations needed by unlearning from the fully trained model and retraining from the initial model to achieve a given excess risk threshold. The bounds would hold for any stochastic, iterative, first-order learning algorithm.

**Theoretical Claims:**

I have checked the correctness of proofs in Appendix A and C.
In the proof of Lemma C.1, the distribution which the expectation is taken w.r.t. should be stated. It is first on the retain set, then on the forget set. if r_f > .5, the bound can just be L.
In the proof of Theorem 2, the connection between Eq (36) and the claim in Theorem 2 is not immediately clear. The readability and clarity of the proofs can be enhanced.

---

> ### Author Rebuttal · Authors · 2025-04-01
>
> We thank *Reviewer tKga* for their in-depth review and detailed analysis of our theoretical work. We address their comments below and will make sure to include the corrected elements into the final version. $$ $$
>
> - **Impossible VS Inefficient Unlearning:** You are correct in saying that unlearning in this regime is inefficient rather than impossible. Therefore, we will rename IR to "Inefficient Unlearning" in the final manuscript. $$ $$
>
> - **Form of the loss function:** The loss function we rely on is inspired from Eq. 26 in order to represent a loss of the "worst-case" type we study in our theoretical analysis. However, it is adapted in order to be both harder (non-smooth) and yield more stable results, as explained in details in our response to *Reviewer KpLi*. $$ $$
>
> - **Experiments on Real Datasets:**
> Following the reviewer’s suggestion, we conducted additional experiments using the standard [real-world dataset "Digits" of handwritten numbers](https://scikit-learn.org/1.5/auto_examples/datasets/plot_digits_last_image.html) to further validate our theoretical findings. We are pleased to report that the three regimes predicted by our theory can be observed in these experiments, as illustrated [here (anonymized link)](https://anonymous.4open.science/r/Submission6456/figure.pdf). Additional experimental details are provided in the second part of our response to Reviewer KpLi. $$ $$
>
> - **Non Standard Definitions:** Our first unlearning definition (Definition 1) is indeed worded differently from other works, but remains close to what other papers in the literature have used, see e.g. [Georgiev et al](https://arxiv.org/pdf/2410.23232)'s Definition 2, [Chourasia et al](https://arxiv.org/pdf/2210.08911)'s Definition 2.4 where the forget set is taken as any subset of the overall dataset. While our Definition 2 is indeed unusual, it relies on the same principles and implicitly uses the concept of neighboring dataset: If $\mathcal{D}_f$ and $\mathcal{D}_f'$ are of size $k$, then the total datasets $\mathcal{D}_r \cup \mathcal{D}_f$ and $\mathcal{D}_r \cup \mathcal{D}_f'$ are $k$-neighbors. It is common for the overall datasets to be neighbors in unlearning definitions, but this is usually not the case for forget sets. $$ $$
>
> - **Out-of-distribution data handling:** In our analysis, there indeed is no direct analysis of the in- vs out-of-distribution character of the data. We do not make specific assumptions on the data as to the distribution they follow. $$ $$
>
> - **Verification and explanation of Ass. 2:** Assumption 2 is a technical assumption needed for the proofs. We believe it could be lifted with little change in term of results but the analysis would be made significantly heavier.
> There are two principal elements to Assumption 2: the existence of $A$ such that $\mathbb{P}(\xi_r \in A) = p$ where $\xi_r \sim \mathcal{D}_r$, and the existence of another data distribution $\mathcal{D}_f'$. The second point is relatively trivial if, as stated in the article,  "$supp(\mathcal{D}_r)$ and $supp(\mathcal{D}_f)$ do not cover the whole space. The first point is a consequence of the Intermediate Value Theorem: Define $f(r) = \mathbb{P}(X \in B(0,r))$. By the continuity of the density, $f$ is continuous, with $f(0)=0$ and $lim f(r)=1$. By the Intermediate Value Theorem, for every $p\in [0,1]$ there exists an $r_p \ge 0$ such that $f(r_p)=p$.
>
> - **Lemma C.1:** We are grateful for the reviewer’s attention to detail and will add the missing expectations, that indeed hold on the retain set, then forget set, respectively.

---

### Official Review · Reviewer_qtfk · 2025-03-14

**Overall Recommendation:** 4

**Summary:**

In this paper, the author revisits the machine unlearning topic and analyzes the efficiency of unlearning methods. The author defines a complexity ratio metric that compares the computation costs of the unlearning method and full model retraining. A phase diagram is drawn based on the complexity ratio and reveals three distinct regimes, which are impossible unlearning, efficient unlearning, and trivial unlearning, respectively. Based on the phase diagram, it answers the question about which would be the most efficient way to achieve the unlearning purpose under certain conditions.

**Claims And Evidence:**

Yes

**Essential References Not Discussed:**

Yes

**Experimental Designs Or Analyses:**

Yes

**Methods And Evaluation Criteria:**

Yes

**Other Comments Or Suggestions:**

In the introduction section, the author mentioned “we provide the first lower bound for the unlearning complexity ratio, answering an open problem in the literature (Allouah et al., 2024)…”. It would be better to explicitly describe what the open problem is.

**Other Strengths And Weaknesses:**

Strength:
* This paper is well-organized.
* The author summarizes many interesting and meaningful insights, which are helpful and can be considered as guidelines for future research in the field of machine unlearning.
* The authors provide extensive theoretical evidence to support their claims.

Weakness:
* In the introduction section, the author defined TS as the stochastic gradient steps required to unlearn by retraining from scratch. But the definition of TU is missing. And here the author defines the T as the steps, where the next sentence says the ratio of time required. Note that the ratio of the number of steps and the ratio of time required is different.
* The author does not explain what the excess risk e is.
* It would be better if the author could have a more comprehensive discussion on the limitations of the proposed theory.
* The evaluation part lacks some practical cases to prove the feasibility of the proposed theory.

**Questions For Authors:**

Please refer to the weakness part.

**Relation To Broader Scientific Literature:**

The author summarizes many interesting and meaningful insights, which are helpful and can be considered as guidelines for future research in the field of machine unlearning.

**Theoretical Claims:**

Yes

---

> ### Author Rebuttal · Authors · 2025-04-01
>
> We thank *Reviewer qtfk* for their positive review of our work and their attention to rigour. Here are our answers to their main comments. $$ $$
>
> - Missing introduction of quantities: We thank the reviewer for pointing out two omissions that affect the clarity of the paper. We appreciate their careful reading and the opportunity to address these issues. $T^U_e$ and $T^S_e$ will be properly introduced, as well as $e$, by this modification of lines 50 to 55:
> > "Specifically, in a general machine learning setting, let us define $r_f$ as the fraction of data in the forget set, and $e$ as the excess risk—that is, the gap in loss between a learned model and the optimal one. We denote by $T^U_e$ (resp. $T^S_e$) the time required for an algorithm to unlearn the forget set (resp. to retrain from scratch without the forget set) until the excess risk falls below $e$. Note that *time* is here measured by the number of accesses to gradients of individual data points, and thus assumes that gradient computation is the main computation bottleneck."
>
> $$ $$
>
> - **Limitations of the proposed theory:** Our theoretical analysis is limited in its scope, as discussed in the conclusion, by the assumptions we make: strong-convexity of the objective, first-order of the algorithms, dependence only on the retain set of the unlearning algorithms, and verification of Assumption 2. Additional limitations include the tightness of our analysis since our bound from Theorem 3 scales with $\sqrt{d}$, which is not matched by our lower bound in Theorem 2, as discussed in subsection 4.5, and the potential non-optimality of Alg. 2, as discussed in subsection 5.2. The reviewer is right in pointing-out that these points are dispersed throughout the paper and we will make sure to group them in our final manuscript.
> $$ $$
>
> - **Experiments on Real Datasets:**
> Following the reviewer’s suggestion, we conducted additional experiments using the standard [real-world dataset "Digits" of handwritten numbers](https://scikit-learn.org/1.5/auto_examples/datasets/plot_digits_last_image.html) to further validate our theoretical findings. We are pleased to report that the three regimes predicted by our theory can be observed in these experiments, as illustrated [here (anonymized link)](https://anonymous.4open.science/r/Submission6456/figure.pdf). Additional experimental details are provided in the second part of our response to *Reviewer KpLi*.

---

### Official Review · Reviewer_KpLi · 2025-03-15

**Overall Recommendation:** 3

**Summary:**

This paper proposes a novel efficiency-oriented metric for machine unlearning. Based on the training resources required for a retrained model, it measures the efficiency of machine unlearning with respect to the size of the forget set and introduces three different regimes. The first regime addresses cases where unlearning can be achieved with minimal cost, while the second and third regimes deal with the lower and upper bounds of the complexity ratio, respectively. The paper empirically demonstrates the validity of these regimes through experiments.

**Claims And Evidence:**

It is positive that the authors clearly present their claims through Assumptions, Theorems, and Corollaries. However, the empirical evidence supporting the existence of such regimes does not appear to be sufficient. The paper only provides experiments on a limited set of settings, and it does not demonstrate how generally these regimes hold.

**Essential References Not Discussed:**

N/A

**Experimental Designs Or Analyses:**

The argument on this point does not seem very convincing to me. I hope the authors will provide a thorough explanation in the rebuttal as to why the experiments presented in the paper are sufficient.

**Methods And Evaluation Criteria:**

As I mentioned earlier, I do not think the paper has conducted an adequate evaluation on this point. However, I am not fully confident in this judgment. To me, this paper appears to be more theory-focused rather than practical, and since I am more of a practitioner than a theorist, it is possible that my evaluation is biased due to my limited perspective. Therefore, I plan to further assess this aspect through discussions with other reviewers during the discussion period.

**Other Comments Or Suggestions:**

The strength of the paper is that it is well-written and easy to follow, while the weakness is that the experiments are not sufficiently comprehensive. Please refer to the above sections for more detailed comments.

**Other Strengths And Weaknesses:**

The strength of the paper is that it is well-written and easy to understand, while the weakness is that the experiments are not sufficiently thorough. Please refer to the above comments for more details.

**Questions For Authors:**

It seems necessary for the authors to provide an explanation of why the current evaluation in the paper is sufficient. As it stands, I am not fully convinced, and therefore, I find it difficult to recommend acceptance at this point.

**Relation To Broader Scientific Literature:**

The paper appears to focus primarily on reviewing DP-based unlearning methods, and I am not sure if this is sufficient. It would have been better if the authors had also mentioned machine unlearning literature from a more practical perspective. However, since the main contribution of this paper lies in providing a theoretical foundation, I consider this a minor concern.

**Theoretical Claims:**

The paper presents theoretical theorems for the three regimes. Although I have read through the logical arguments, I have not verified them rigorously.

---

> ### Author Rebuttal · Authors · 2025-04-01
>
> We thank *Reviewer KpLi* for their insights and appreciate their practice-oriented review of our paper. We think adding the following discussion to the final version of the paper will benefit it greatly. $$  $$
>
> - **Choice of experiments:** Our work studies the ratio of "worst case" unlearning and retraining time, meaning that the we bound the convergence time for every algorithm when optimising over the hardest possible functions. Our theoretical analysis is based on "worst-case" functions that have forms such as the one described in Eq. 26, used to prove lower bounds on convergence speeds: $\ell^g(\boldsymbol{\theta}, \xi) = \frac{\mu}{2} \|\boldsymbol{\theta}\|^2 - \frac{L}{2} g(\xi) \theta_1$.
> Therefore, in order to assess the relevance and significance of our theoretical analysis, it makes sense to empirically evaluate how fast unlearning and retraining algorithms go when optimising functions of this form.
> We started by optimizing over this loss function directly, but faced two issues.
> The first and most significant was that we did not observe the linear delimitation between TR and ER theorized in Theorem 1, the second was numerical instability. The first problem occurred because of the smoothness of the function: since Theorem 1 does not assume smoothness of the loss and Eq 26. provides a smooth loss, TR expanded beyond what was expected, forming a square root boundary rather than a linear one.
> For this reason, we introduced non-smoothness in the form of an $L1$ penalty, making the loss: $\ell^g(\boldsymbol{\theta}, \xi) = \frac{\mu}{2} \|\boldsymbol{\theta}\|^2 - \frac{L}{2} g(\xi) \theta_1 + \lambda \|\boldsymbol{\theta}_{2:d}\|_1$, which fixed the issue and provided a loss function better-aligned with the theory. The second issue was numerical instability: since every parameter but the first one was quickly decreasing to $0$, the remaining task was optimising over the first parameter. The first parameter naturally oscillated between values over and under the optimal one, resulting in a noisy process that could converge prematurely by chance. Therefore, we chose to expand the loss on the first parameter to half of the model's parameters, solving the second issue, and making the loss
>
> $$\ell^g(\boldsymbol{\theta}, \xi) = \frac{\mu}{2} \left\|\boldsymbol{\theta}\right\|^2 - \frac{L}{4} g(\xi) \sum_{i=1}^{d/2} \theta_i + \frac{L}{4} \sum_{i=d/2 + 1}^{d} \left|\theta_i\right|.$$ While uncommon, it represents a hard loss function that verifies every assumption made in the paper, which is why we chose it to perform our experiments.
> $$ $$
>
> - **Practical applicability of the experiments:** In order to asses the practical applicability of our theory, we corroborated our experimental results with additional experiments on [the real-world dataset "Digits" from sklearn](https://scikit-learn.org/1.5/auto_examples/datasets/plot_digits_last_image.html). We performed a logistic regression on this well-known dataset of handwritten digits. We used a standard cross-entropy loss with $L2$ weight regularisation (thus ensuring strong convexity and Lipschitzness of the loss) to train a linear classifier on the data, and compare unlearning and retraining times, as done in Figure 2. [The result for the experiment is available at this anonymised link](https://anonymous.4open.science/r/Submission6456/figure.pdf), in compliance with the [ICML 2025 Peer Review FAQ](https://icml.cc/Conferences/2025/PeerReviewFAQ). Additionally, since adding this new experiment broadens the scope of the experimental section and further verifies our theoretical findings, we will include them in the updated version of the paper. The specific hyperparameters used to generate this figure are: batch size of $64$, use of SGD with a starting learning rate of $0.01$, decaying by a factor of $0.6$ every $1000$ batch, with a penalty weight of $1$ and results averaged over $50$ seeds.
> $$ $$
>
> - **Mention of non DP-based unlearning methods:** While we do mention several unlearning methods  with both no unlearning guarantees and with non DP-based guarantees in the first and second paragraphs of our literature review, we will include an additional discussion on the link between these papers and our, as explained in more details in our response to *Reviewer ak4V*.

---

> > ### Comment · Reviewer_KpLi · 2025-04-02
> >
> > I am satisfied with the author response and will finalize my score as a 3.

---

### Official Review · Reviewer_ak4V · 2025-03-16

**Overall Recommendation:** 4

**Summary:**

In this paper, the authors studied the complexity trade-offs of machine unlearning, in particular, at which conditions, unlearning can outperform retraining from scratch regarding the efficiency or unlearning cannot bring any benefits. To accomplish this, the authors proposed to leverage a phase diagram to reveal these regions, which is supported by the authors' rigorous theoretical analysis throughout the paper.



## update after rebuttal
I went over the authors' responses and other reviewers' comments. I think the authors have adequately addressed all of our comments, in particular, our concerns about the real datasets. Since all of us lean to accept this paper, I would love to increase the score to 4.

**Claims And Evidence:**

Most claims of the paper look fine to me.

**Essential References Not Discussed:**

I think the authors primarily focus on one type of machine unlearning approach. It would be better if the authors could incorporate some discussion in the paper to discuss the connections of the proposed approach to other machine unlearning approaches, in particular, those not dependent on adding gaussian noises, say Bourtoule, Lucas, Varun Chandrasekaran, Christopher A. Choquette-Choo, Hengrui Jia, Adelin Travers, Baiwu Zhang, David Lie, and Nicolas Papernot. "Machine unlearning." In 2021 IEEE symposium on security and privacy (SP), pp. 141-159. IEEE, 2021.  and Wu, Yinjun, Edgar Dobriban, and Susan Davidson. "Deltagrad: Rapid retraining of machine learning models." In International Conference on Machine Learning, pp. 10355-10366. PMLR, 2020.

**Experimental Designs Or Analyses:**

Yes, I checked the experimental designs and analyses. I feel that the authors may need additional experiments on real datasets to justify the theoretical results in this paper.

**Methods And Evaluation Criteria:**

Yes, the proposed methods and evaluation make sense to me.

**Other Comments Or Suggestions:**

See above comments

**Other Strengths And Weaknesses:**

Strengths:
+ The authors have done a series of theoretical analyses to reveal the complexity trade-offs in machine unlearning, which is perfect.
+ The authors' overall presentation is clear and easy to follow
+ The authors also performed some empirical studies, which can justify the phase diagram proposed by them.

Weakness:
- One of my biggest concerns is that the analysis conducted in this paper is closely tied to Algorithm 2. I am not sure whether the analysis could apply to general scenarios. Maybe I missed something but it would be great if the authors could have some discussion on this.
- I think it would be better if the authors could perform experiments on real datasets rather than synthetic ones to verify their findings. If the results on real datasets match their analysis, that would be more convincing
- I also feel a little confused about Theorem 3. Based on my understanding, this theorem should correspond to the ER area in Figure 1, right? According to Figure 1, the ER area only appears between $a * \frac{e}{e_0}$ and $b * \frac{e}{e_0}$ where $0 < a < b < 1$ for any fixed $k_{\epsilon, \delta}$. But it seems to me that Theorem 3 applied to any $e$ smaller than $e_0$. Although the authors attempt to have this discussion in paragraphs after Theorem 3, I still feel that more clarification would be essential (perhaps by combining Figure 1).

**Questions For Authors:**

See above comments

**Relation To Broader Scientific Literature:**

The contributions of this paper are closely related to the area of machine unlearning.

**Theoretical Claims:**

The theoretical claims and analysis look fine to me.

---

> ### Author Rebuttal · Authors · 2025-04-01
>
> We thank *Reviewer ak4V* for their detailed feedback on our work, and we especially appreciate their focus on the theoretical elements of the paper. Here is a point-by-point response to their questions: $$ $$
>
> - **Link to non DP-based unlearning papers:** There are indeed several ways of achieved certified unlearning besides DP-based MU methods. While we have mentioned some of them in the literature review, including *Bourtoule et al*’s "Machine Unlearning" paper, it is true that our work would benefit from discussing these elements in more details later in the analysis. Indeed, there is no direct equivalence between DP-based guarantees and the type of guarantees offered by order two methods like "Deltagrad" or exact unlearning methods like SISA (from *Bourtoule*’s work).
> However, our impossibility result in Theorem 3 has interesting consequences as to the feasibility of exact unlearning in our setup. Since the "unlearn" and "retrain" distributions ought to be the exact same in exact unlearning, it implies (0,0)-Unlearning as defined in Definition 1. Thus, $\kappa_{\epsilon,\delta}=\infty$ and Theorem 2 implies that $T^U_e$ cannot asymptotically improve over $T^S_e$, regardless of $e$. Note that this result does not contradict the potential efficiency of exact unlearning methods since they usually rely on a modification of the training process, which is outside of the theoretical framework laid-out in this paper.
> As for the link between second-order and first-order unlearning, since the unlearning guarantees used are very different by nature, they are hardly comparable theoretically. However, auditing first and second-order MU methods and their relative resilience to various attacks would be a valuable research direction and is left to future works. $$ $$
>
>
> - **Role played by Algorithm 2**: The exact role that Algorithm 2 holds in our paper is indeed a subtle question, that we are happy to expand upon. Please be assured that the following elements will be included in our final version.
> In our analysis, we rely on Alg. 2 solely in order to show the existence of efficient first-order unlearning algorithms, but our analysis characterises the behaviour of every first-order algorithm as long as it relies only on the retain set. Indeed, Alg. 2 is only used in the proof of Theorem 3 to characterize the regime in which we know that unlearning algorithms can outperform retraining asymptotically. Therefore, the boundaries of the IR and TR are independent from Alg. 2, which is only used in Theorem 3 to describe the ER. $$ $$
>
>
> - **Theorem 3 and the Efficient Regime:** We thank the reviewer for pointing out a potential source of confusion for readers, as Theorem 3 is in essence not a result that delimits the efficient regime, but rather shows that efficient methods do exist in this regime. In particular, Theorem 3 only provides the unlearning time for Alg. 2, which happens to improve upon retraining from scratch precisely in the efficient regime. To improve readability, we have decided to rename Theorem 3 as *"noise and fine tune efficiency"*, and add the following corollary to this theorem:
> **Corollary (Efficient regime).** There exists a universal constant $c > 0$ such
> that, for any $\gamma\in(0,1)$, if $e \geq \frac{c}{\gamma} \left(\frac{r_f}{1-r_f}\right)^2 \left(1+d\kappa_{\varepsilon,\delta}^2\right) e_0$, then
> $$
> \frac{T^U_e}{T^S_e} < \gamma.
> $$
>
>
> - **Experiments on Real Datasets:**
> Following the reviewer’s suggestion, we conducted additional experiments using the standard [real-world dataset "Digits" of handwritten numbers](https://scikit-learn.org/1.5/auto_examples/datasets/plot_digits_last_image.html) to further validate our theoretical findings. We are pleased to report that the three regimes predicted by our theory can be observed in these experiments, as illustrated [here (anonymized link)](https://anonymous.4open.science/r/Submission6456/figure.pdf). Additional experimental details are provided in the second part of our response to *Reviewer KpLi*.

---

> > ### Comment · Reviewer_ak4V · 2025-04-05
> >
> > I would love to thank the authors for their responses. I think they have addressed my concerns. I would love to wait for other reviewers' further comments. If there is no serious issue, I would love to increase my score. Thanks!

---

### Decision · Program_Chairs · 2025-05-01

**Decision:**

Accept (poster)

**Comment:**

Overall, I think this is a solid contribution with the following strengths:
- The paper considers a specific form of machine unlearning algorithm that does not require access to the forget set, and establishes asymptotic conditions under which unlearning can be trivial, efficient or inefficient relative to the re-training (from scratch) unlearning scheme.
- The paper validates this theoretical finding on a couple of real datasets with a loss function that satisfies the necessary assumptions.

Overall, this is an interesting paper that tries to provide deeper insights into the area of machine unlearning. The reviewers raised some valid concerns which the authors appeared to satisfactorily address in the author responses. One limitation is that this form of analysis is specific to a specific class of unlearning algorithms, and it does not say much about other forms of unlearning algorithms. However, the authors do clearly acknowledge this limitations.